# Approximation to Smooth Functions by Low-Rank Swish Networks

**Zimeng Li**[1]  **Hongjun Li**[2]  **Jingyuan Wang**[1 3 4]  **Ke Tang**[2]

## Abstract

While deep learning has witnessed remarkable achievements in a wide range of applications, its substantial computational cost imposes limitations on application scenarios of neural networks. To alleviate this problem, low-rank compression is proposed as a class of efficient and hardware-friendly network compression methods, which reduce computation by replacing large matrices in neural networks with products of two small ones. In this paper, we implement low-rank networks by inserting a sufficiently narrow linear layer without bias between each of two adjacent nonlinear layers. We prove that low-rank Swish networks with a fixed depth are capable of approximating any function from the Hölder ball $\mathcal{C}^{\beta,R}([0,1]^d)$ within an arbitrarily small error where $\beta$ is the smooth parameter and $R$ is the radius. Our proposed constructive approximation ensures that the width of linear hidden layers required for approximation is no more than one-third of the width of nonlinear layers, which implies that the computational cost can be decreased by at least one-third compared with a network with the same depth and width of nonlinear layers but without narrow linear hidden layers. Our theoretical finding can offer a theoretical basis for low-rank compression from the perspective of universal approximation theory.

## 1. Introduction

The universal approximation theory (UAT) for neural networks mainly studies the quantitative and qualitative aspects

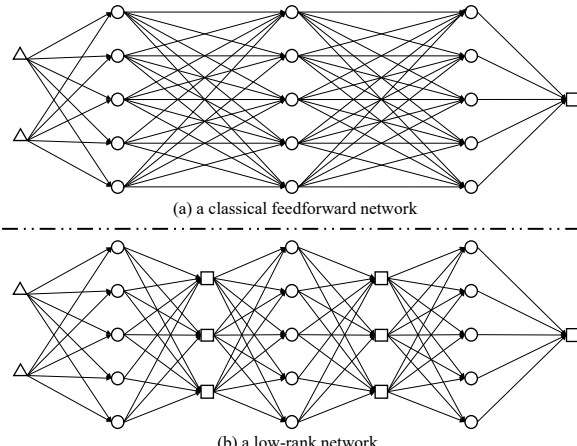

*Figure 1.* An illustration of the difference between classical feedforward network and low-rank network. A classical feedforward network is composed of several nonlinear hidden layers and a linear output layer. A low-rank network is composed of several interleaved nonlinear and linear layers where the last linear layer act as the output layer. A "△" stands for an input neuron, a "◯" stands for a nonlinear neuron (i.e. neuron with activation function), and a "□" stands for a linear neuron (i.e. neuron without activation function).

of how neural networks can approximate a specific class of functions to an arbitrarily small error (Hornik et al., 1989; Cybenko, 1989; Leshno et al., 1993; Yarotsky, 2017; Yarotsky & Zhevnerchuk, 2020; Siegel, 2023; Li et al., 2024a), providing a solid foundation for understanding their outstanding performance in a wide variety of fields such as computer vision (Tolstikhin et al., 2021; Li et al., 2024b), speech recognition (Graves et al., 2013; Ren et al., 2019), natural language processing (Touvron et al., 2023; Li et al., 2023), intelligent healthcare (Ren et al., 2021; Shi et al., 2022), and smart city (Wang et al., 2022a;b; Jiang et al., 2023; Ji et al., 2025).

In the realm of theoretical statistics, the upper bound of the approximation rate is one of the most important ingredients to derive the consistency and convergence rate of neural network estimates (Chen & White, 1999; Schmidt-Hieber, 2020; Kohler & Langer, 2021; Farrell et al., 2021). Based on

[1]School of Computer Science and Engineering, Beihang University, Beijing, China [2]Institute of Economics (School of Social Sciences), Tsinghua University, Beijing, China [3]School of Economics and Management, Beihang University, Beijing, China [4]Engineering Research Center of Advanced Computer Application Technology, Ministry of Education, China. Correspondence to: Hongjun Li <hongjunli@tsinghua.edu.cn>, Jingyuan Wang <jywang@buaa.edu.cn>.

*Proceedings of the 42nd International Conference on Machine Learning*, Vancouver, Canada. PMLR 267, 2025. Copyright 2025 by the author(s).

the convergence rate, we could further obtain the pointwise asymptotic normality of neural network estimates and the asymptotic normality of functionals of neural network estimates, which is a key step towards constructing confidence intervals and conducting hypothesis testings (Shintani & Linton, 2004; Horel & Giesecke, 2020; Zhong et al., 2022). Evidently, the UAT for neural networks occupies a central position within the framework of learning theory.

Numerous works focused on UATs of ReLU neural networks(Yarotsky, 2017; 2018; Opschoor et al., 2022), owing to its highly efficient computation. However, due to the fact that the derivative of ReLU is zero on $(-\infty, 0)$, if the input of a ReLU neuron is less than zero, its related parameters cannot be trained, which is referred to as the "dying ReLU" problem. Moreover, ReLU suffers from poor smoothness, as it merely possesses a discontinuous weak derivative up to the first order and the higher order derivatives are zero almost everywhere. Consequently, ReLU neural networks are unable to achieve universal approximation with higher-order Sobolev norms. In practice, problems in some fields such as ecology, economics, and engineering physics concern not only the estimation of the unknown function $f_0$, but also the estimation of its high-order derivatives(Shyu & Caswell, 2014; Wang & Werning, 2022), which cannot be estimated using ReLU networks.

The Swish (SiLU, sigmoid-weighted linear unit) activation function [1] inherits the advantages of ReLU while alleviating the above problems, since Swish is an infinitely differentiable function with a shape similar to that of ReLU and a nonzero derivative almost everywhere. Empirical studies across various tasks and architectures show that Swish neural networks generally perform better than ReLU neural networks and are seldom significantly inferior to other popular activation functions such as ELU, GELU, Swish, and Mish(Eger et al., 2018; Dubey et al., 2022). However, works on UATs related to Swish neural networks are scarce.

Network compression aims to decrease the computational and memory costs of neural networks by compressing their sizes. Common network compression methods can be classified into four categories: pruning (Dong et al., 2017), low-rank compression (Idelbayev & Carreira-Perpiñán, 2020), quantization (Jacob et al., 2018), and knowledge distillation (Hinton et al., 2015). Except low-rank compression, the remaining categories of methods are all underpinned by UATs to some extent. For pruning, Yarotsky(2017), Petersen & Voigtlaender(2018), and Bolcskei et al.(2019) show that sparse neural networks, which can be viewed as the results of pruning fully-connected networks, could also

be universal approximators. Their results indicate that only $\mathcal{O}(L\mathcal{H})$ nonzero parameters are required to achieve the optimal approximation rate, where $L$ is the depth and $\mathcal{H}$ is the width. For quantization, Petersen & Voigtlaender(2018) proves that networks with parameters encoded by $\mathcal{O}(\log_2 \frac{1}{\varepsilon})$ bits can approximate piecewise smooth functions to $\varepsilon$ and Gühring & Raslan(2021) proves the same quantization condition for approximating smooth functions by networks with general smooth activation functions. As for knowledge distillation methods, all research efforts involving upper and lower bounds of sizes of networks needed to achieve a rapid approximation rate can offer valuable insights for the design of student networks (Shen et al., 2022b; Hon & Yang, 2022; Liu & Chen, 2024).

Low-rank compression is a class of efficient and hardware-friendly neural network compression techniques that approximate weight matrices through matrix factorization (Denil et al., 2013; Sainath et al., 2013). A standard low-rank compression pipeline typically involves two key steps: first, decomposing weight matrices of the trained network into pairs of low-rank matrices, followed by fine-tuning the resulting low-rank network on the training dataset. Intuitively, whether low-rank compression can preserve performance without significant degradation depends on the ranks of the original weight matrices, which are inherently shaped by the training data distribution. Consequently, there is no universal guarantee that low-rank compression will remain effective across diverse tasks and domains. However, extensive empirical evidence suggests that low-rank compression can achieve significant computational savings while maintaining nearly identical network performance in most practical applications. Our main theoretical result (Theorem 4.1) provides a principled explanation for the universal effectiveness of low-rank compression in preserving model performance.

In this paper, we develop the theoretical foundation for low-rank compression from the perspective of approximation theory by answering the question of whether a low-rank neural network can serve as a good approximator for a wide range of functions. To be specific, we consider how Swish neural networks with a sufficiently narrow linear layer without bias between each of two adjacent nonlinear layers, called low-rank Swish networks, can approximate any function from the Hölder ball $\mathcal{C}^{\beta,R}([0,1]^d)$ where $\beta \in \mathbb{R}_+$ is the smooth parameter and $R \in \mathbb{R}_+$ is the radius of the ball. For any $f \in C^{\beta,R}([0,1]^d)$, we divide $[0,1]^d$ into $M^d$ hypercubes where $M \in \mathbb{N}_+$, then approximate $f$ by a sum–product combination of Taylor expansions and approximate bump functions [2] at all grid points of $[0,1]^d$ where an approximate bump function at a point refers to a scalar function whose absolute value is small when the

---

[1] Strictly speaking, Swish $x \mapsto x(1 + e^{-\beta x})^{-1}$ proposed by Ramachandran et al.(2018) and SiLU $x \mapsto x(1 + e^{-x})^{-1}$ proposed by Elfwing et al.(2018) are slightly different. This paper ignored the difference by defaulting $\beta$ to 1.

---

[2] Here we adopt the term "approximate bump function" to distinguish from "bump function" (also called "test function") which refers to infinitely differentiable function with compact support.

input is away from the point. Then we construct Taylor polynomials and approximate bump functions using neural networks respectively, multiply them together, and sum up.

Our main contributions are as follows:

- We derive an upper bound of error for approximating any function from the Hölder ball $\mathcal{C}^{\beta,R}([0,1]^d)$ by using low-rank Swish networks and provide the required depth, width of linear hidden layers, width of nonlinear layers, upper bound of number of nonzero parameters, and upper bound of absolute values of parameters.

- Our constructive approximation guarantees that the width of linear hidden layers is no more than one-third of the width of nonlinear layers, indicating the quantity of multiplication operations occurred in all hidden layers except the first one could be reduced by at least one-third compared with a network with the same depth and width of nonlinear layers but without linear hidden layers.

## 2. Related Works

### 2.1. Universal Approximation

The research on UATs began with one-hidden-layer neural networks. Hornik et al.(1989) proved that one-hidden-layer neural networks activated by an arbitrary squashing function are capable of approximating any measurable function on a compact set to any small error measured in the sup norm. In the same year, a similar result was published by Cybenko(1989). Hornik et al.(1990) improved Hornik et al.(1989)'s result by replacing the sup norm with the first-order Sobolev norm. Barron(1993) further specified that the approximation rate of one-hidden-layer neural networks for a specific class of functions is of the order $\mathcal{O}(\frac{1}{n})$ where $n$ represents the number of hidden nodes.

With the development of computational technology, the training and deployment of deep neural networks have become possible. A series of works have demonstrated that for certain functions, if approximated by shallow networks, the required width is far greater than that needed when approximated by deep networks (Eldan & Shamir, 2016; Telgarsky, 2016; Safran & Shamir, 2017; Rolnick & Tegmark, 2018).

In the last decade, works on approximation theories of deep ReLU networks account for a large proportion. Yarotsky(2017) first demonstrated how to approximate general smooth functions using deep ReLU networks. He proved that ReLU networks with depth $\mathcal{O}(\log(\frac{1}{\varepsilon}))$ and number of nonzero parameters $\mathcal{O}(\varepsilon^{-\frac{d}{n}}\log(\frac{1}{\varepsilon}))$ can approximate any function from the unit ball of Sobolev space $\mathcal{W}^{n,\infty}([0,1]^d)$ within $\varepsilon$. Yarotsky(2018) studied approximations of continuous functions on compact domains by deep ReLU networks.

Liu & Chen(2024) proved that deep ReLU networks with width $d+1$ can achieve the optimal approximation rate where $d$ is the input dimension. DeVore et al.(2021) wrote a survey on UATs of ReLU networks.

Deep neural networks activated by popular ReLU-like functions, including ELU (Clevert et al., 2016), GELU (Hendrycks & Gimpel, 2016), Swish (Ramachandran et al., 2018), and Mish (Misra, 2020), have attained great empirical success in a diverse range of real-world applications (Kenton & Toutanova, 2019; Bochkovskiy et al., 2020), inspiring theoretical exploration of networks activated by them. Ohn & Kim(2019) achieved an approximation theorem appropriate for deep neural networks activated by a wide range of functions. Although their result encompassed Swish neural networks, we demonstrate in Corollary 4.2 that the same approximation error can be achieved with a shallower depth and a smaller upper bound of absolute values of parameters. Zhang et al.(2024) showed that ReLU networks can be approximated by networks with commonly used activation functions, at the cost of only increasing the depth and width of the networks by a small constant multiple.

Besides classical feedforward networks, there are some novel studies on the approximation capabilities of modern network architectures. Shen et al.(2022a) derived a non-asymptotic approximation error bound for deep convolutional neural networks in Sobolev space. Yun et al.(2020) and Zaheer et al.(2020) showed transformers are universal approximators of sequence-to-sequence functions. Lin & Jegelka(2018) studied the universal approximation property of residual networks activated by ReLU.

### 2.2. Approximation Theory Foundations for Network Compression

Network compression methods are divided into four major categories: pruning, low-rank compression, quantization, and knowledge distillation. Apart from low-rank compression, all the others have evidence from the universal approximation theory to support their rationality.

Pruning methods downsize a network by eliminating either unimportant parameters (LeCun et al., 1989; Dong et al., 2017) or (groups of) neurons (Xia et al., 2022; Ko et al., 2023). The former is generally referred to as unstructured pruning, while the latter is known as structured pruning. For unstructured pruning, Yarotsky(2017), Petersen & Voigtlaender(2018), and Bolcskei et al.(2019) showed sparse neural networks are enough to achieve the optimal approximation rate for smooth functions with $\mathcal{O}(L\mathcal{H})$ nonzero parameters where $L$ represents the depth and $\mathcal{H}$ represents the width. The UAT foundation of structured pruning is presented together with knowledge distillation later.

Quantization methods are designed to cut down on both

computational load and storage requirements by employing a reduced number of bits to encode parameters(Jacob et al., 2018). Petersen & Voigtlaender(2018) demonstrated that networks with parameters encoded using $\mathcal{O}(\log_2 \frac{1}{\varepsilon})$ bits are capable of approximating piecewise smooth functions with error $\varepsilon$. Similarly, Gühring & Raslan (2021) established the identical quantization condition for approximating smooth functions by networks with general smooth activation functions.

Knowledge distillation utilizes the outputs of a large network as labels to train a small network, then replaces the large network with the small one to achieve compression (Hinton et al., 2015; Mirzadeh et al., 2020; Kim et al., 2022). In terms of compression, both knowledge distillation and structured pruning employ a network that is narrower and/or shallower to substitute for the original network. Research efforts on the upper and lower bounds of network sizes for rapid approximation rates can offer theoretically guaranteed structural design for compressed networks (Shen et al., 2022b; Hon & Yang, 2022; Liu & Chen, 2024).

# 3. Preliminaries

## 3.1. Notations

We denote the set of real numbers by $\mathbb{R}$, the set of positive real numbers by $\mathbb{R}_+$, the set of natural numbers by $\mathbb{N}$, and $\mathbb{N} - \{0\}$ by $\mathbb{N}_+$. For $n \in \mathbb{N}$, we denote the set $\{0, 1, \ldots, n\}$ by $[n]$ and $[n] - \{0\}$ by $[n]_+$. If $n = 0$, then $[n]_+$ is $\emptyset$. For $x \in \mathbb{R}$, $\lceil x \rceil$ denotes the smallest integer greater than or equal to $x$ and $\lfloor x \rfloor$ denotes the largest integer less than or equal to $x$.

Vectors are denoted by bold lowercase letters, for example $\boldsymbol{x} := (x_1, x_2, \ldots, x_d)^\top \in \mathbb{R}^d$ is a $d$-dimensional vector. Matrices are denoted by bold uppercase letters, for example $\boldsymbol{W} \in \mathbb{R}^{m \times n}$ is a matrix with $m$ rows and $n$ columns whose element at $i$-th row and $j$-th column is $w_{ij}$. A $d$-dimensional multi-index $\boldsymbol{\alpha}$ is a vector in $\mathbb{N}^d$. For a multi-index $\boldsymbol{\alpha}$ and a vector $\boldsymbol{x}$, we denote $|\boldsymbol{\alpha}| := \sum_{i=1}^{d} \alpha_i$, $\boldsymbol{\alpha}! := \prod_{i=1}^{d} \alpha_i$, and $\boldsymbol{x}^{\boldsymbol{\alpha}} := \prod_{i=1}^{d} x_i^{\alpha_i}$.

Let $\mathscr{X}$ and $\mathscr{Y}$ be two sets. The notation $f : \mathscr{X} \to \mathscr{Y}$ denotes the function $f$ with domain $\mathscr{X}$ and co-domain $\mathscr{Y}$. For an univariate function $f : \mathscr{X} \subset \mathbb{R} \to \mathbb{R}$, its $n$-th derivative is denoted by $f^{(n)}$. If $n \leq 3$, we also use $f'$, $f''$, and $f'''$ to denote 1-st, 2-nd, and 3-rd derivatives respectively. For a multivariate function $f : \mathscr{X} \subset \mathbb{R}^d \to \mathbb{R}$ and a multi-index $\boldsymbol{\alpha}$, we denote

$$\partial^{\boldsymbol{\alpha}} := \frac{\partial^{|\boldsymbol{\alpha}|}}{\partial^{\alpha_1} \partial^{\alpha_2} \ldots \partial^{\alpha_d}}$$

and the $\boldsymbol{\alpha}$-order partial derivative of $f$ by $\partial^{\boldsymbol{\alpha}} f$.

The meaning of sup norm $\| \cdot \|_\infty$ varies with its input. For a vector $\boldsymbol{x}$, $\|\boldsymbol{x}\|_\infty := \max_i |x_i|$. For a matrix $\boldsymbol{A}$, $\|\boldsymbol{A}\|_\infty :=$ $\max_{i,j} |a_{i,j}|$. For a function $f : \mathscr{X} \to \mathbb{R}$, $\|f\|_\infty := \sup_{x \in \mathscr{X}} |f(x)|$. Note that for a function $f : \mathscr{X} \to \mathbb{R}^m$ with integer $m > 1$, $\|f(\boldsymbol{x})\|_\infty := \max_i |(f(\boldsymbol{x}))_i|$ because $f(\boldsymbol{x})$ is a vector. For a vector or matrix, $\| \cdot \|_0$ denotes its total number of nonzero elements. Finally, we denote the combinatorial number for all $m, n \in \mathbb{N}$ by

$$\binom{m}{n} := \begin{cases} \frac{m!}{n!(m-n)!}, & m \geq n \geq 0, \\ 0, & \text{otherwise.} \end{cases}$$

## 3.2. Low-Rank Swish Network

Drawing upon established works in low-rank compression (including but not limited to Denil et al.(2013), Sainath et al.(2013), and Idelbayev & Carreira-Perpiñán(2020)), we formally define low-rank network as:

**Definition 3.1** (Low-rank Network). Let $d, o \in \mathbb{N}_+$ be the input and output dimensions and $\rho : \mathbb{R} \to \mathbb{R}$ be the nonlinear activation function. For a vector input $\boldsymbol{x} \in \mathbb{R}^n$, $\rho(\boldsymbol{x}) := (\rho(x_1), \rho(x_2), \ldots, \rho(x_n))^\top$. Let $L$ be the depth, i.e. the number of nonlinear layers, $\mathcal{H}$ be the width of nonlinear layers and $H_i$ be the width of $i$-th linear hidden layer such that

$$2H < \mathcal{H} \tag{1}$$

that is called the low-rank condition. A low-rank Swish network $nn : \mathscr{X} \subset \mathbb{R}^d \to \mathbb{R}^o$ with depth $L$, width of nonlinear layers $\mathcal{H}$, width of linear hidden layers $H$, number of nonzero parameters $S$, and maximum absolute value of parameters $B$ is a function defined by

$$nn(\boldsymbol{x}) := l_L \circ l_{L-1} \circ \cdots \circ l_2 \circ l_1(\boldsymbol{x}) + \boldsymbol{b}_{L+1} \tag{2}$$

$$l_i(\boldsymbol{z}) := V_i \rho(\boldsymbol{W}_i \boldsymbol{z} + \boldsymbol{b}_i) \quad (i \in [L]_+) \tag{3}$$

where $\boldsymbol{W}_1 \in \mathbb{R}^{\mathcal{H} \times d}$, $\boldsymbol{W}_i \in \mathbb{R}^{\mathcal{H} \times H}$ for $i \in [L]_+ - \{1\}$, $\boldsymbol{V}_i \in \mathbb{R}^{H \times \mathcal{H}}$ for $i \in [L-1]_+$, $\boldsymbol{V}_L \in \mathbb{R}^{o \times \mathcal{H}}$, $\boldsymbol{b}_i \in \mathbb{R}^{\mathcal{H}}$ for $i \in [L]_+$, and $\boldsymbol{b}_{L+1} \in \mathbb{R}^o$ such that

1. $S = \sum_{i=1}^{L}(\|\boldsymbol{W}_i\|_0 + \|\boldsymbol{V}_i\|_0) + \sum_{i=1}^{L+1} \|\boldsymbol{b}_i\|_0$,

2. $B = \max\{\max_{i \in [L]_+} \|\boldsymbol{W}_i\|_\infty, \max_{i \in [L]_+} \|\boldsymbol{V}_i\|_\infty, \max_{i \in [L+1]_+} \|\boldsymbol{b}_i\|_\infty\}$.

Figure 1 illustrates the difference between the classical feedforward network and the low-rank network. Here we explain why the "low-rank" network achieves low-rank compression. Let's denote the output of the $i$-th nonlinear layer by $\boldsymbol{z}_i \in \mathbb{R}^{\mathcal{H}}$ and the output of the $i + 1$-th nonlinear layer by $\boldsymbol{z}_{i+1} \in \mathbb{R}^{\mathcal{H}}$. In a low-rank network, by Definition 3.1, we have

$$\boldsymbol{z}_{i+1} = \rho(\boldsymbol{W}_{i+1} \boldsymbol{V}_i \boldsymbol{z}_i + \boldsymbol{b}_{i+1}) \tag{4}$$

where the weight $\boldsymbol{W}_{i+1} \in \mathbb{R}^{\mathcal{H} \times H}$ and the weight $\boldsymbol{V}_i \in \mathbb{R}^{H \times \mathcal{H}}$. For a classical feedforward network with the same depth and width of nonlinear layers,

$$\boldsymbol{z}_{i+1} = \rho(\widetilde{\boldsymbol{W}}_{i+1} \boldsymbol{z}_i + \widetilde{\boldsymbol{b}}_{i+1}) \tag{5}$$

where the weight $\widetilde{\boldsymbol{W}}_{i+1} \in \mathbb{R}^{\mathcal{H} \times \mathcal{H}}$. The low-rank condition (1) ensures that the number of elements in $\boldsymbol{W}_{i+1}$ and $\boldsymbol{V}_i$ is no more than the number of elements in $\widetilde{\boldsymbol{W}}_{i+1}$, i.e.

$$2H\mathcal{H} < \mathcal{H}^2, \tag{6}$$

suggesting the memory required for storing weight matrices is compressed. And the condition (1) also naturally implies that the quantity of multiplication operations to calculate from $\boldsymbol{z}_i$ to $\boldsymbol{z}_{i+1}$ in the low-rank network is no more than that in the classical feedforward network, i.e.

$$H\mathcal{H}^2 + \mathcal{H}H^2 < \mathcal{H}^3, \tag{7}$$

suggesting the computational cost is compressed.

At the end, the Swish activation function $\rho : \mathbb{R} \to \mathbb{R}$ is defined by

$$\rho(x) := \frac{x}{1 + e^{-x}}. \tag{8}$$

A low-rank network activated by Swish is called a low-rank Swish network.

### 3.3. Hölder Function

**Definition 3.2** (Hölder Space). Let $d \in \mathbb{N}_+$, $\mathscr{X} \in \mathbb{R}^d$ and $\beta \in \mathbb{R}_+$. There exist $\kappa \in \mathbb{N}$ and $0 < \gamma \le 1$ such that $\beta = \kappa + \gamma$. For a function $f : \mathscr{X} \to \mathbb{R}$, its Hölder norm is defined by

$$\|f\|_{\mathcal{C}^\beta} := \max \left\{ \sup_{|\boldsymbol{\alpha}| \le \kappa} \|\partial^{\boldsymbol{\alpha}} f\|_\infty, \right.$$
$$\left. \sup_{|\boldsymbol{\alpha}| = \kappa} \sup_{\boldsymbol{x} \ne \boldsymbol{y}} \frac{|\partial^{\boldsymbol{\alpha}} f(\boldsymbol{x}) - \partial^{\boldsymbol{\alpha}} f(\boldsymbol{y})|}{\|\boldsymbol{x} - \boldsymbol{y}\|_\infty^\gamma} \right\} \tag{9}$$

And the Hölder space $\mathcal{C}^\beta([0,1]^d)$ is defined as the set

$$\left\{ f : \mathscr{X} \to \mathbb{R} \mid \|f\|_{\mathcal{C}^\beta} < \infty \right\} \tag{10}$$

equipped with Hölder norm $\| \cdot \|_{\mathcal{C}^\beta}$.

We call functions in $\mathcal{C}^\beta([0,1]^d)$ Hölder functions. When $0 < \beta \le 1$ (i.e. $\kappa = 0$), we call them Hölder continuous functions. When $\beta > 1$ (i.e. $\kappa \in \mathbb{N}_+$), we call them Hölder smooth functions. Next we define the Hölder ball with radius $R$.

**Definition 3.3** (Hölder Ball). Let $d \in \mathbb{N}_+$, $\mathscr{X} \in \mathbb{R}^d$, $R \in \mathbb{R}_+$, and $\beta = \kappa + \gamma, \kappa \in \mathbb{N}, \gamma \in (0,1]$. The Hölder ball $\mathcal{C}^{\beta,R}([0,1]^d)$ is defined by

$$\left\{ f : \mathscr{X} \to \mathbb{R} \mid \|f\|_{\mathcal{C}^\beta} \le R \right\}. \tag{11}$$

## 4. Approximation Theorem for Low-Rank Swish Neural Networks

**Theorem 4.1.** *Let* $\beta \in \mathbb{R}_+, \beta = \kappa + \gamma, \kappa \in \mathbb{N}, \gamma \in (0,1]$, *and* $R \in \mathbb{R}_+$. *For all* $f \in \mathcal{C}^{\beta,R}([0,1]^d)$, $M \in \mathbb{N}_+$, $\lambda \ge$

$2^{-\frac{1}{3}}$, *and* $\tau \ge 1$, *there exists a low-rank Swish network* $nn : [0,1]^d \to \mathbb{R}$ *with depth*

$$\max \left\{ \left\lceil \frac{\kappa}{2} \right\rceil, \lceil \log_2 d \rceil + 1 \right\} + 1,$$

*width of nonlinear layers*

$$2 \binom{d+1}{d-1} + 4 \binom{d+\kappa-2}{d-1} + 4 \binom{d+\kappa-1}{d-1} + 6(M+1)^d,$$

*width of linear hidden layers*

$$\binom{d+1}{d-1} + \binom{d+\kappa-3}{d-1} + \binom{d+\kappa-2}{d-1} + 2(M+1)^d,$$

*upper bound of absolute values of parameters*

$$\max \left\{ (3M+2)\tau, 2\lambda^2 \max_{|\boldsymbol{\alpha}| \le \kappa} \left\{ \sum_{\substack{\boldsymbol{\nu} \ge \boldsymbol{\alpha} \\ |\boldsymbol{\nu}| \le \kappa}} \frac{R}{\boldsymbol{\nu}!} \prod_{i=1}^d \binom{\nu_i}{\alpha_i} \right\}, 2\lambda^2 \right\},$$

*and upper bound of number of nonzero parameters*

$$c_1 + c_2(M+1)^d$$

*such that*

$$|nn(\boldsymbol{x}) - f(\boldsymbol{x})|$$
$$\le c_3 \frac{(M+1)^d}{\lambda^2} + c_4 M^{-\beta} + c_5(M+1)^d \tau e^{-\tau} \tag{12}$$

*for all* $\boldsymbol{x} \in [0,1]^d$, *where* $c_1, c_2, c_3, c_4$, *and* $c_5$ *are positive constants depending only on* $d, \kappa$, *and* $R$.

An extended version of Theorem 4.1 is presented in Appendix A as Theorem A.24. In Theorem A.24 we provide the exact formulas for upper bounds of the number of nonzero parameters and the approximation error. Next, we show a way to set the network size in Corollary 4.2 to ensure that the approximation error can be arbitrarily small.

**Corollary 4.2.** *Let* $\beta > 0$ *and* $R \in \mathbb{R}_+$. *For all* $0 < \varepsilon \le 3c_4$, *there exists a low-rank Swish network* $nn : [0,1]^d \to \mathbb{R}$ *with depth* $\mathcal{O}(1)$, *width of nonlinear layers* $\mathcal{O}(\varepsilon^{-\frac{d}{\beta}})$, *width of linear hidden layers* $\mathcal{O}(\varepsilon^{-\frac{d}{\beta}})$, *maximum absolute value of parameters* $\mathcal{O}(\varepsilon^{-\frac{\beta+d}{\beta}})$, *and number of nonzero parameters* $\mathcal{O}(\varepsilon^{-\frac{d}{\beta}})$ *such that*

$$|f(\boldsymbol{x}) - nn(\boldsymbol{x})| \le \varepsilon \qquad \forall \boldsymbol{x} \in [0,1]^d. \tag{13}$$

The proof ideas of Theorem 4.1 is showed in section 5. And the rigorous proofs of Theorem 4.1 and Corollary 4.2 are showed in Appendix A.

*Remark* 4.3 (Comparison with (Ohn & Kim, 2019)). Ohn & Kim(2019) demonstrated in their Theorem 1 that for any continuous piecewise linear or locally quadratic function there exists a feedforward neural network activated by it with depth $\mathcal{O}(\log \frac{1}{\varepsilon})$, width $\mathcal{O}(\varepsilon^{-\frac{d}{\beta}})$, number of nonzero paramters $\mathcal{O}(\varepsilon^{-\frac{d}{\beta}} \log(\frac{1}{\varepsilon}))$, and maximum absolute value of parameters $\mathcal{O}(\varepsilon^{-\frac{4(\beta+d)}{\beta}})$ can approximate any functions from the Hölder ball $\mathcal{C}^{\beta,R}([0,1]^d)$ within $\varepsilon$. Because the Swish is a locally quadratic function, their result holds for Swish networks. Compared with Corollary 4.2, we only requires a constant depth to achieve an approximation error within $\varepsilon$. Moreover, our growth rates of the number of nonzero parameters and the maximum absolute value of parameters with respect to $\varepsilon$ are better than those in Theorem 1 in (Ohn & Kim, 2019).

*Remark* 4.4 (Low-rank compression). In Theorem 4.1, we notice that, when $\beta > 2$ (i.e. $\kappa \geq 2$), the width of linear hidden layers is always no more than one-third of the width of nonlinear layers, since

$$3\left(\binom{d+1}{d-1} + \binom{d+\kappa-3}{d-1} + \binom{d+\kappa-2}{d-1} + 2(M+1)^d\right)$$

$$\leq 2\binom{d+1}{d-1} + 4\binom{d+\kappa-2}{d-1} + 4\binom{d+\kappa-1}{d-1} + 6(M+1)^d$$

$$\Leftarrow \binom{d+1}{d-1} \leq \binom{d+\kappa-2}{d-1} + \binom{d+\kappa-1}{d-1}$$

$$\Leftarrow \binom{d+1}{d-1} \leq \binom{d}{d-1} + \binom{d+1}{d-1}.$$

For a low-rank network with depth $L$, width of linear hidden layers $H$, and width of nonlinear layers $\mathcal{H}$, excluding the first and the last layers, evaluating it at one point requires $(L-1)(H\mathcal{H}^2 + H^2\mathcal{H})$ multiplication operations. However, for a classical feedforward network with the same depth and width of nonlinear layers, excluding the first and the last layers, evaluating it at one point requires $(L-1)\mathcal{H}^3$ multiplication operations. When $\mathcal{H} \geq 3H$, the low-rank network can guarantee that the quantity of multiplication operations is reduced by at least one-third compared to the classical feedforward network, since

$$(L-1)(H\mathcal{H}^2 + H^2\mathcal{H}) \leq \frac{2}{3}(L-1)\mathcal{H}^3$$

$$\Leftarrow H\mathcal{H}^2 + H^2\mathcal{H} \leq \frac{2}{3}\mathcal{H}^3$$

$$\Leftarrow \frac{\mathcal{H}^3}{3} + \frac{\mathcal{H}^3}{9} \leq \frac{2}{3}\mathcal{H}^3.$$

*Remark* 4.5 (Curse of dimensionality). In the realm of neural network approximation theory, the curse of dimensionality refers to the phenomenon that as the input dimension $d$ goes to infinity, the network size required to achieve a given approximation error grows fast or the approximation error grows fast when the network size is fixed. Corollary 4.2 implies that our approximation result suffers from the curse

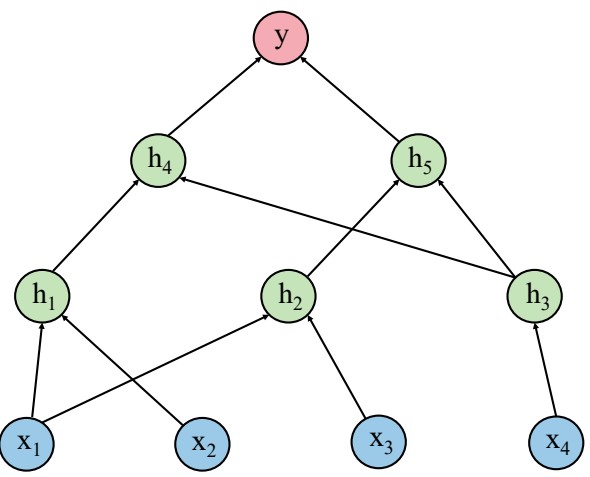

*Figure 2.* An illustration of a hierarchical composite function. $x_1$, $x_2$, $x_3$, and $x_4$ are input variables. $h_1 = h_1(x_1, x_2)$, $h_2 = h_2(x_1, x_3)$, $h_3 = h_3(x_4)$, $h_4 = h_4(h_1, h_3)$, $h_5 = h_5(h_2, h_3)$, and $y = y(h_4, h_5)$. Though the input dimension of the hierarchical composite function is 4, the input dimensions of its component functions do not exceed 2.

of dimensionality, which poses difficulties in approximating high-dimensional functions. Here we briefly introduce a class of high-dimensional functions, called hierarchical composite functions, which are universal in reality and can be approximated without being affected by the curse of dimensionality. A hierarchical composite function, as shown in Figure 2, is composed of multiple layers of functions, and each component function is of low input dimension. It is obvious that the network size required to approximate a hierarchical composite function is directly related to the input dimension of each component function and has no direct relation to the input dimension of the hierarchical composite function, because we can construct networks to approximate component functions respectively, then combine them into one network (Schmidt-Hieber, 2020; Kohler & Langer, 2021).

## 5. Proof Ideas

The proof of Theorem 4.1 can be segmented into four steps:

**Step 1: approximating any Hölder function $f$ by a sum–product combination of Taylor polynomials and approximate bump functions**

Let $M \in \mathbb{N}_+$. We divide $[0,1]^d$ into congruent hypercubes with side length $1/M$, then get $(M+1)^d$ grid points. For any $f \in \mathcal{C}^{\beta,R}([0,1]^d)$ and any $\boldsymbol{m} \in [M]^d$, its $\kappa$-order Taylor expansion at the grid point $\boldsymbol{m}/M$ is denoted by

$$P_{\boldsymbol{m}}^{\kappa}(\boldsymbol{x}) := \sum_{|\boldsymbol{\alpha}| \leq \kappa} \frac{\partial^{\boldsymbol{\alpha}} f(\boldsymbol{m}/M)}{\boldsymbol{\alpha}!} \left(\boldsymbol{x} - \frac{\boldsymbol{m}}{M}\right)^{\boldsymbol{\alpha}}, \quad (14)$$

where $\boldsymbol{\alpha} \in \mathbb{N}^d$ stands for multi-index.

**Lemma 5.1.** *Let* $\beta \in \mathbb{R}_+, \beta = \kappa + \gamma, \kappa \in \mathbb{N}, \gamma \in (0, 1]$, *and* $R \in \mathbb{R}_+$. *For all* $f \in \mathcal{C}^{\beta, R}([0, 1]^d), M \in \mathbb{N}_+, \boldsymbol{m} \in [M]^d$, *and* $\boldsymbol{x} \in [0, 1]^d$,

$$|f(\boldsymbol{x}) - P_{\boldsymbol{m}}^\kappa(\boldsymbol{x})| \leq \binom{\kappa + d - 1}{d - 1} R \left\| \boldsymbol{x} - \frac{\boldsymbol{m}}{M} \right\|_\infty^\beta. \quad (15)$$

Lemma 5.1 shows $P_{\boldsymbol{m}}^\kappa$ is a good approximator around $\boldsymbol{m}/M$, but the error cannot be well controlled when $\boldsymbol{x}$ is far away from $\boldsymbol{m}/M$. Next we introduce a technique proposed by (Gühring & Raslan, 2021) to deal with this problem.

Let $\tau \in \mathbb{R}_+$. Define $\phi_{\boldsymbol{m}}^\tau$, an approximate bump function at the grid point $\boldsymbol{m}/M$, by

$$\psi^\tau(x) := \frac{1}{\tau}(\rho(\tau(x + 2)) - \rho(\tau(x + 1))$$
$$- \rho(\tau(x - 1)) + \rho(\tau(x - 2))) \quad (16)$$

$$\phi_{\boldsymbol{m}}^\tau(\boldsymbol{x}) := \prod_{i=1}^d \psi^\tau \left( 3M \left( x_i - \frac{m_i}{M} \right) \right). \quad (17)$$

The graph of $\phi_{\boldsymbol{m}}^\tau$ looks like a bump at the grid point $\boldsymbol{m}/M$. Lemma A.7 guarantees that $|\phi_{\boldsymbol{m}}^\tau(\boldsymbol{x})|$ is bounded when $\|\boldsymbol{x} - \boldsymbol{m}/M\|_\infty \geq 1/M$ and the bound goes to zero as $\tau$ increases. By intuition, $\phi_{\boldsymbol{m}}^\tau$ can preserve the value of $P_{\boldsymbol{m}}^\kappa$ when $\boldsymbol{x}$ is near $\boldsymbol{m}/M$ and eliminate the influence of $P_{\boldsymbol{m}}^\kappa$ when $\boldsymbol{x}$ is far from $\boldsymbol{m}/M$. Thus, for every grid point, we use the product of the Taylor polynomial and the approximate bump function at this point to approximate $f$ around this point, then sum up the products at all grid points to approximate $f$ on $[0, 1]^d$.

**Lemma 5.2.** *Let* $\beta > 0, \beta = \kappa + \gamma, \kappa \in \mathbb{N}, \gamma \in (0, 1]$, *and* $R \in \mathbb{R}_+$. *For all* $f \in \mathcal{C}^{\beta, R}([0, 1]^d), M \in \mathbb{N}_+, \tau \geq 1$ *and* $\boldsymbol{x} \in [0, 1]^d$,

$$\left| f(\boldsymbol{x}) - \sum_{\boldsymbol{m} \in [M]^d} P_{\boldsymbol{m}}^\kappa(\boldsymbol{x}) \phi_{\boldsymbol{m}}^\tau(\boldsymbol{x}) \right|$$
$$\leq 6\tau e^{-\tau} R \frac{(2\|\rho'\|_\infty)^d - 1}{2\|\rho'\|_\infty - 1} +$$
$$3^d M^{-\beta} \binom{\kappa + d - 1}{d - 1} R (2\|\rho'\|_\infty)^d + \quad (18)$$
$$6(M + 1)^d \tau e^{-\tau} \binom{\kappa + d - 1}{d - 1} R (2\|\rho'\|_\infty)^{d-1}.$$

**Step 2: approximating** $(P_{\boldsymbol{m}}^\kappa)_{\boldsymbol{m} \in [M]^d}$ **by a low-rank Swish network** $\mathcal{P}$

**Lemma 5.3.** *Let* $\boldsymbol{a} \in \mathbb{R}^d$ *and* $b_{\boldsymbol{\alpha}} \in \mathbb{R}$ *for all* $\boldsymbol{\alpha} \in \mathbb{N}^d$ *with* $|\boldsymbol{\alpha}| \leq \kappa$. *For all* $\boldsymbol{x} \in \mathbb{R}^d$,

$$\sum_{|\boldsymbol{\alpha}| \leq \kappa} b_{\boldsymbol{\alpha}} (\boldsymbol{x} - \boldsymbol{a})^{\boldsymbol{\alpha}} = \sum_{|\boldsymbol{\alpha}| \leq \kappa} \boldsymbol{x}^{\boldsymbol{\alpha}} \sum_{\substack{\boldsymbol{\nu} \geq \boldsymbol{\alpha} \\ |\boldsymbol{\nu}| \leq \kappa}} b_{\boldsymbol{\nu}} \prod_{i=1}^d \binom{\nu_i}{\alpha_i} (-a_i)^{\nu_i - \alpha_i}. \quad (19)$$

Note that for two multi-indexes we say $\boldsymbol{\nu} \geq \boldsymbol{\alpha}$ iff $\nu_i \geq \alpha_i$ for all $i$. Lemma 5.3 shows $P_{\boldsymbol{m}}^\kappa$ could be represented as a linear combination of monomials $\boldsymbol{x}^{\boldsymbol{\alpha}}$ with $|\boldsymbol{\alpha}| \leq \kappa$. Next we construct all monomials $\boldsymbol{x}^{\boldsymbol{\alpha}}$ in a network and Taylor polynomials at all grid points by linear combinations of $\boldsymbol{x}^{\boldsymbol{\alpha}}$ whose coefficients determined by Lemma 5.3.

First we show a Swish network of depth 1 and width 2 can approximate the square function.

**Lemma 5.4.** *Let* $\lambda > 0$. *Then for all* $x \in \mathbb{R}$, *there exist* $\xi$ *between* 0 *and* $\frac{x}{\lambda}$ *and* $\zeta$ *between* 0 *and* $-\frac{x}{\lambda}$ *such that*

$$2\lambda^2 \left( \rho\left(\frac{x}{\lambda}\right) + \rho\left(-\frac{x}{\lambda}\right) \right) = x^2 + \frac{\rho^{(4)}(\xi) + \rho^{(4)}(\zeta)}{12} \cdot \frac{x^4}{\lambda^2} \quad (20)$$

*and*

$$\left| 2\lambda^2 \left( \rho\left(\frac{x}{\lambda}\right) + \rho\left(-\frac{x}{\lambda}\right) \right) - x^2 \right| \leq \frac{x^4}{12\lambda^2}. \quad (21)$$

Together with the polarization identity, we show a Swish network of depth 1 and width 4 can approximate the multiplication function.

**Lemma 5.5.** *Let* $\lambda > 0$. *For all* $x, y \in \mathbb{R}$,

$$\left| 2\lambda^2 \left( \rho\left(\frac{x + y}{2\lambda}\right) + \rho\left(-\frac{x + y}{2\lambda}\right) - \rho\left(\frac{x - y}{2\lambda}\right) - \right.\right.$$
$$\left.\left. \rho\left(-\frac{x - y}{2\lambda}\right) \right) - xy \right| \leq \frac{1}{12\lambda^2} \cdot \frac{x^4 + 6x^2y^2 + y^4}{8}. \quad (22)$$

Next we show a Swish network of depth 1 and width 2 can mimic the identity function exactly.

**Lemma 5.6.** *For all* $x \in \mathbb{R}$,

$$\rho(x) - \rho(-x) = \frac{x}{1 + e^{-x}} - \frac{-x}{1 + e^x} = x. \quad (23)$$

Next we briefly describe how to construct monomials and Taylor polynomials in a network as depicted in Figure 3 in Appendix A. The detailed construction is presented in the proof of Lemma A.17. According to Lemma 5.6, 5.4 and 5.5, we use a nonlinear layer followed by a linear layer to construct all 1st- and 2nd-order monomials from input variables $x_1, x_2, \ldots, x_d$. Then we utilize another linear layer of width $(M + 1)^d$ linked to the previous nonlinear layer to construct the first two orders of Taylor polynomials at all $(M+1)^d$ grid points. The weights and biases are determined by Lemma 5.6, 5.4, 5.5, and 5.3. We concatenate these two linear layers in parallel as one which follows behind the nonlinear layer. Next, following the similar way, we construct all 2nd-, 3rd-, and 4th-order monomials from 1st- and 2nd-order monomials via a nonlinear layer followed by a linear layer. And we use another nonlinear layer followed by a linear layer to preserve the first two orders of all Taylor

polynomials by Lemma 5.6, then connect this linear layer to the previous nonlinear layer which constructs 3rd- and 4th-order monomials to approximate the first four orders of all Taylor polynomials. Then we concatenate these two nonlinear layers and two linear layers respectively as one nonlinear layer followed by one linear layer. In the following steps, letting the initial value of $l$ be 2, we repeat the process until $(P_{\boldsymbol{m}}^{\kappa})_{\boldsymbol{m}\in[M]^d}$ is completely constructed:

1. using a nonlinear layer followed by a linear layer to construct 2nd-, $(2l+1)$th-, and $(2l+2)$th-order monomials from 2nd-, $(2l-1)$th-, and $(2l)$th-order monomials;

2. using another nonlinear layer followed by a linear layer to preserve the first $2l$ orders of $(P_{\boldsymbol{m}}^{\kappa})_{\boldsymbol{m}\in[M]^d}$, then adding connections to the nonlinear layer which constructs $(2l+1)$th- and $(2l+2)$th-order monomials to approximate the first $2l+2$ orders of $(P_{\boldsymbol{m}}^{\kappa})_{\boldsymbol{m}\in[M]^d}$;

3. concatenating these two nonlinear layers and two linear layers in parallel respectively as one nonlinear layer followed by one linear layer;

4. letting $l := l + 2$ and constructing the next nonlinear and linear layers by step 1 to 4 until $(P_{\boldsymbol{m}}^{\kappa})_{\boldsymbol{m}\in[M]^d}$ is completely constructed.

We denote the network constructed above by $\mathcal{P} : [0,1]^d \to \mathbb{R}^{(M+1)^d}$. The approximation error and network size is shown in Lemma A.17.

**Step 3: approximating $(\phi_{\boldsymbol{m}}^{\tau})_{\boldsymbol{m}\in[M]^d}$ by a low-rank Swish network $\mathcal{G}$**

It is obvious that $\psi^{\tau}(3M(x_i - \frac{m_i}{M}))$ can be exactly constructed by a nonlinear layer followed by a linear layer. Then, to construct $\phi_{\boldsymbol{m}}^{\tau}$, the key is to construct the product of $d$ variables. For convience, we suppose $d = 2^q$ where $q \in \mathbb{N}$ and denote $\psi^{\tau}(3M(x_i - \frac{m_i}{M}))$ as $z_i$. We approximate the mapping $(z_1, z_2, \ldots, z_d) \mapsto (z_1 z_2, z_3 z_4, \ldots, z_{2^q-1} z_{2^q})^{\top}$ using a nonlinear layer followed by a linear layer according to Lemma 5.5. By applying the above way $q$ times iteratively, we get $\prod_{i=1}^{d} z_i$, i.e. $\phi_{\boldsymbol{m}}^{\tau}$. For all $\boldsymbol{m} \in [M]^d$, we construct $\phi_{\boldsymbol{m}}^{\tau}$ in parallel. We denote the network constructed above by $\mathcal{G} : [0,1]^d \to \mathbb{R}^{(M+1)^d}$. The approximation error and network size is shown in Lemma A.22.

**Step 4: approximating $\sum_{\boldsymbol{m}\in[M]^d} P_{\boldsymbol{m}}^{\kappa}\phi_{\boldsymbol{m}}^{\tau}$ by the inner product of $\mathcal{P}$ and $\mathcal{G}$**

Based on the constructive approximation before, we have that network $\mathcal{P}$ approximates $(P_{\boldsymbol{m}}^{\kappa})_{\boldsymbol{m}\in[M]^d}$ and network $\mathcal{G}$ approximates $(\phi_{\boldsymbol{m}}^{\tau})_{\boldsymbol{m}\in[M]^d}$. Considering that the depths of $\mathcal{P}$ and $\mathcal{G}$ may be different, we construct several nonlinear and linear layers according to Lemma 5.6 to align their depths. And we still denote the two aligned networks by

*Table 1.* Cross-validation results for classical feedforward networks and low-rank networks on various classification (top) and regression (bottom) datasets. $L$ represents the depth (i.e. the number of nonlinear layers) of both networks and $\mathcal{H}$ represents the width of nonlinear layers of both networks.

| DATASET | $L$ | $\mathcal{H}$ | ACC(%) | | t-statistic |
|---|---|---|---|---|---|
| | | | classical | low-rank | |
| Iris | 4 | 20 | $95.3 \pm 4.3$ | $94.7 \pm 5.0$ | 0.36 |
| Rice | 2 | 35 | $92.7 \pm 1.9$ | $92.6 \pm 2.0$ | 1.00 |
| BankMarketing | 2 | 188 | $68.9 \pm 15.3$ | $71.1 \pm 15.4$ | $-2.01$ |
| Adult | 2 | 540 | $85.8 \pm 0.3$ | $85.8 \pm 0.3$ | $-0.47$ |

| DATASET | $L$ | $\mathcal{H}$ | RMSE | | t-statistic |
|---|---|---|---|---|---|
| | | | classical | low-rank | |
| RealEstate | 4 | 30 | $.078 \pm .021$ | $.077 \pm .020$ | 1.29 |
| Abalone | 3 | 50 | $.077 \pm .022$ | $.077 \pm .022$ | $-0.44$ |
| WineQuality | 4 | 78 | $.123 \pm .009$ | $.123 \pm .009$ | 1.21 |
| BikeSharing | 4 | 60 | $.100 \pm .036$ | $.070 \pm .024$ | 3.90 |

$\mathcal{P}$ and $\mathcal{G}$. By Lemma 5.5, we construct a nonlinear layer with width $4(M+1)^d$ and a subsequent linear layer with width 1 to multiply the output dimensions of $\mathcal{P}$ and $\mathcal{G}$ corresponding to the same grid point respectively, and then sum them up. We denote the final network by $nn$ which approximates $\sum_{\boldsymbol{m}\in[M]^d} P_{\boldsymbol{m}}^{\kappa}(\boldsymbol{x})\phi_{\boldsymbol{m}}^{\tau}(\boldsymbol{x})$. The approximation error and size of $nn$ is showed in Theorem 4.1 and Theorem A.24.

# 6. Experiments

Our Theorem 4.1 shows that for a classical feedforward Swish network with appropriate size, compressing each of its weight matrix of size $\mathcal{H} \times \mathcal{H}$ to the product of two small matrices of size $\mathcal{H} \times \frac{\mathcal{H}}{3}$ and $\frac{\mathcal{H}}{3} \times \mathcal{H}$ will not result in a loss of approximation ability. Here we conduct experiments to verify that the ratio $1/3$ is safe.

We choose eight popular UCI datasets, four of which are used for classification tasks and four for regression tasks. For each dataset, we convert each category feature to several dummy features, then scale all features to $[0,1]$. For regression datasets, we also scale the targets to $[0,1]$. Table 2 in Appendix B records the basic information for these datasets. Then, for each dataset, we employ grid search with 10-fold cross-validation to identify the optimal depth and width for the classical feedforward Swish network. The candidate set for the depth consists of $\{2, 3, 4\}$, and for the width, it is $\{4d, 5d, 6d\}$, where $d$ represents the input dimension. Subsequently, we conduct 10-fold cross-validation to evaluate the classical feedforward Swish network of the optimal depth and width and the low-rank Swish network whose depth and width of nonlinear layers are the same as those of the classical feedforward Swish network and width of

linear hidden layers is one-third of that of nonlinear layers. In addition, we perform dependent t-tests for paired samples on the cross-validation results.

The results of t-tests in Table 1 indicate that on all datasets, classical feedforward Swish networks do not significantly outperform low-rank Swish networks. Conversely, on the BikeSharing dataset, the root mean square error (RMSE) of the classical feedforward Swish network is significantly higher than that of the low-rank Swish network. The experimental results indicate that the compression ratio of $1/3$ suggested by our Theorem 4.1 is reliable.

## 7. Conclusion

In this paper, we establish the theoretical foundation for low-rank compression from the perspective of universal approximation theory. Specifically, we prove that for any Hölder function, there exists a Swish network with narrow linear hidden layers sandwiched between adjacent nonlinear layers, which can approximate the Hölder function within a given small error. Through our constructive approximation, we find that the width of the linear hidden layers is at most one-third of that of the nonlinear layers. This leads to a significant reduction: the number of multiplication operations occurring in all hidden layers except the first one can be decreased by at least one-third compared with a classical feedforward network having the same depth and width of nonlinear layers. Extensive experiments have confirmed the reliability of our theoretical result. This research not only enriches the theoretical understanding of low-rank compression but also holds great potential for practical applications where computational efficiency is crucial.

## Acknowledgements

Jingyuan Wang acknowledges the financial support of National Natural Science Foundation of China (No. 72222022, 72171013). Hongjun Li acknowledges the financial support of National Natural Science Foundation of China (No. 72342032). Ke Tang acknowledges the financial support of National Natural Science Foundation of China (No. 72192802, 72342008).

## Impact Statement

This paper presents work whose goal is to advance the field of Machine Learning. There are many potential societal consequences of our work, none which we feel must be specifically highlighted here.

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

## A. Technical Proofs

### A.1. Approximatng $f \in \mathcal{C}^{\beta,R}([0,1]^d)$ by $\sum_{\boldsymbol{m}\in[M]^d} P_{\boldsymbol{m}}^{\kappa} \phi_{\boldsymbol{m}}^{\tau}$

First we prove Lemma 5.1 which shows the approximation error of a Taylor polynomial at a grid point.

*Proof of Lemma 5.1.* By Taylor expansion theorem, there exists $\xi_m \in [0,1]$ for all $m \in [M]$ such that $\forall \boldsymbol{x} \in [0,1]^d$,

$$
\begin{aligned}
|f(\boldsymbol{x}) - P_{\boldsymbol{m}}^{\kappa}(\boldsymbol{x})| &= \left| \sum_{|\boldsymbol{\alpha}|=\kappa} \partial^{\boldsymbol{\alpha}} f\left(\frac{\boldsymbol{m}}{M} + \xi_m\left(\boldsymbol{x} - \frac{\boldsymbol{m}}{M}\right)\right) \frac{(x-\boldsymbol{m}/M)^{\boldsymbol{\alpha}}}{\boldsymbol{\alpha}!} - \sum_{|\boldsymbol{\alpha}|=\kappa} \partial^{\boldsymbol{\alpha}} f\left(\frac{\boldsymbol{m}}{M}\right) \frac{(x-\boldsymbol{m}/M)^{\boldsymbol{\alpha}}}{\boldsymbol{\alpha}!} \right| \\
&\leq \sum_{|\boldsymbol{\alpha}|=\kappa} \left| \partial^{\boldsymbol{\alpha}} f\left(\frac{\boldsymbol{m}}{M} + \xi_m\left(\boldsymbol{x} - \frac{\boldsymbol{m}}{M}\right)\right) - \partial^{\boldsymbol{\alpha}} f\left(\frac{\boldsymbol{m}}{M}\right) \right| \cdot \frac{|(\boldsymbol{x}-\boldsymbol{m}/M)^{\boldsymbol{\alpha}}|}{\boldsymbol{\alpha}!} \\
&\leq \sum_{|\boldsymbol{\alpha}|=\kappa} R \left\| \xi_m\left(x - \frac{\boldsymbol{m}}{M}\right) \right\|_{\infty}^{\gamma} \cdot |(\boldsymbol{x}-\boldsymbol{m}/M)^{\boldsymbol{\alpha}}| \qquad \text{(because } f \in \mathcal{C}^{\beta,R}([0,1]^d)) \\
&\leq \sum_{|\boldsymbol{\alpha}|=\kappa} R \left\| \boldsymbol{x} - \frac{\boldsymbol{m}}{M} \right\|_{\infty}^{\beta} \\
&= \binom{\kappa+d-1}{d-1} R \left\| \boldsymbol{x} - \frac{\boldsymbol{m}}{M} \right\|_{\infty}^{\beta} .
\end{aligned}
\tag{24}
$$

$\square$

Next we show the boundedness of $P_{\boldsymbol{m}}^{\kappa}$ which is used to prove Lemma 5.2 latter.

**Lemma A.1** (Boundedness of $P_{\boldsymbol{m}}^{\kappa}$). *For all $\boldsymbol{x} \in [0,1]^d$,*

$$
|P_{\boldsymbol{m}}^{\kappa}(\boldsymbol{x})| \leq \binom{\kappa+d-1}{d-1} R \left\| \boldsymbol{x} - \frac{\boldsymbol{m}}{M} \right\|_{\infty}^{\beta} + R.
\tag{25}
$$

*Proof of Lemma A.1.* For all $\boldsymbol{x} \in [0,1]^d$, by Lemma 5.1,

$$
\begin{aligned}
|P_{\boldsymbol{m}}^{\kappa}(\boldsymbol{x})| &\leq |P_{\boldsymbol{m}}^{\kappa}(\boldsymbol{x}) - f(\boldsymbol{x})| + |f(\boldsymbol{x})| \\
&\leq \binom{\kappa+d-1}{d-1} R \left\| \boldsymbol{x} - \frac{\boldsymbol{m}}{M} \right\|_{\infty}^{\beta} + R.
\end{aligned}
\tag{26}
$$

$\square$

Next we show some important properties of $\phi_{\boldsymbol{m}}^{\tau}$ which are used in the proof of Lemma A.17.

**Lemma A.2.** $\forall x \geq 1$, $|\rho'(x) - 1| \leq 3xe^{-x}$.

*Proof of Lemma A.2.* $\forall x \geq 0$,

$$
\begin{aligned}
|\rho'(x) - 1| &= \left| \frac{1 + e^{-x} + xe^{-x}}{(1+e^{-x})^2} - 1 \right| \\
&= \left| \frac{e^{-x} - xe^{-x} + e^{-2x}}{(1+e^{-x})^2} \right| \\
&\leq e^{-x} + xe^{-x} + e^{-2x} \\
&\leq 3xe^{-x}.
\end{aligned}
$$

$\square$

**Lemma A.3.** $\forall x \leq -1$, $|\rho'(x)| \leq -3xe^{x}$.

*Proof of Lemma A.3.* $\forall x \leq -1$,

$$
\begin{aligned}
|\rho'(x)| &= \left| \frac{1 + e^{-x} + xe^{-x}}{(1 + e^{-x})^2} \right| \\
&\leq \left| \frac{1 + e^{-x} + xe^{-x}}{e^{-2x}} \right| \\
&\leq e^{2x} + e^x - xe^x \\
&\leq -3xe^x.
\end{aligned}
$$

$\square$

**Lemma A.4** (Boundedness of $\phi_{\boldsymbol{m}}^\tau$). *Let $\tau \in \mathbb{R}$. For all $\boldsymbol{x} \in \mathbb{R}^d$,*

$$
|\phi_{\boldsymbol{m}}^\tau(\boldsymbol{x})| \leq (2\|\rho'\|_\infty)^d. \tag{27}
$$

*Proof of Lemma A.4.* For all $\boldsymbol{x} \in \mathbb{R}^d$, by Lagrange's Mean Value Theorem, there exist $\xi_i, \zeta_i \in [1, 2]$ $(i = 1, 2, \ldots, d)$ such that

$$
\begin{aligned}
|\phi_{\boldsymbol{m}}^\tau(\boldsymbol{x})| &= \prod_{i=1}^d \left| \psi^\tau \left( 3M \left( x_i - \frac{m_i}{M} \right) \right) \right| \\
&= \prod_{i=1}^d \left| \rho' \left( 3M\tau \left( x_i - \frac{m_i}{M} \right) + \xi_i \tau \right) - \rho' \left( 3M\tau \left( x_i - \frac{m_i}{M} \right) - \zeta_i \tau \right) \right| \\
&\leq (2\|\rho'\|_\infty)^d.
\end{aligned} \tag{28}
$$

$\square$

**Lemma A.5** (Locality of $\phi_{\boldsymbol{m}}^\tau$. part I). *Let $\tau \geq 1$. If $x_j - \frac{m_j}{M} \geq \frac{1}{M}$ for some $j \in \{1, 2, \ldots, d\}$,*

$$
|\phi_{\boldsymbol{m}}^\tau(\boldsymbol{x})| \leq (2\|\rho'\|_\infty)^{d-1} \cdot 6\tau e^{-\tau}. \tag{29}
$$

*Proof of Lemma A.5.* By Lagrange's Mean Value Theorem, there exist $\xi_i, \zeta_i \in [1, 2]$ $(i = 1, 2, \ldots, d)$ such that

$$
\begin{aligned}
|\phi_{\boldsymbol{m}}^\tau(\boldsymbol{x})| &= \prod_{i=1}^d \left| \psi^\tau \left( 3M \left( x_i - \frac{m_i}{M} \right) \right) \right| \\
&= \prod_{i=1}^d \left| \rho' \left( 3M\tau \left( x_i - \frac{m_i}{M} \right) + \xi_i \tau \right) - \rho' \left( 3M\tau \left( x_i - \frac{m_i}{M} \right) - \zeta_i \tau \right) \right| \\
&\leq (2\|\rho'\|_\infty)^{d-1} \left| \rho' \left( 3M\tau \left( x_j - \frac{m_j}{M} \right) + \xi_j \tau \right) - \rho' \left( 3M\tau \left( x_j - \frac{m_j}{M} \right) - \zeta_j \tau \right) \right| \\
&\leq (2\|\rho'\|_\infty)^{d-1} \left( \left| \rho' \left( 3M\tau \left( x_j - \frac{m_j}{M} \right) + \xi_j \tau \right) - 1 \right| + \left| 1 - \rho' \left( 3M\tau \left( x_j - \frac{m_j}{M} \right) - \zeta_j \tau \right) \right| \right).
\end{aligned} \tag{30}
$$

Because $x_j - \frac{m_j}{M} \geq \frac{1}{M}$ and $\tau \geq 1$, we have

$$
3M\tau \left( x_j - \frac{m_j}{M} \right) + \xi_j \tau \geq 3M\tau \left( x_j - \frac{m_j}{M} \right) - \zeta_j \tau \geq 3\tau - \zeta_j \tau \geq \tau \geq 1. \tag{31}
$$

Together with the fact that $xe^{-x}$ decreases monotonically on $[1, +\infty)$, it follows

$$
\begin{aligned}
|\phi_{\boldsymbol{m}}^\tau(\boldsymbol{x})| &\leq (2\|\rho'\|_\infty)^{d-1} \left( \left| \rho' \left( 3M\tau \left( x_j - \frac{m_j}{M} \right) + \xi_j \tau \right) - 1 \right| + \left| 1 - \rho' \left( 3M\tau \left( x_j - \frac{m_j}{M} \right) - \zeta_j \tau \right) \right| \right) \\
&\leq (2\|\rho'\|_\infty)^{d-1} \left( 3 \left( 3M\tau \left( x_j - \frac{m_j}{M} \right) + \xi_j \tau \right) e^{-\left( 3M\tau \left( x_j - \frac{m_j}{M} \right) + \xi_j \tau \right)} + \right. \\
&\quad \left. 3 \left( 3M\tau \left( x_j - \frac{m_j}{M} \right) - \zeta_j \tau \right) e^{-\left( 3M\tau \left( x_j - \frac{m_j}{M} \right) - \zeta_j \tau \right)} \right) \qquad \text{(by Lemma A.2)} \\
&\leq (2\|\rho'\|_\infty)^{d-1} \cdot 6\tau e^{-\tau}.
\end{aligned} \tag{32}
$$

$\square$

**Lemma A.6** (Locality of $\phi_{\boldsymbol{m}}^{\tau}$. part II)**.** *Let $\tau \geq 1$. If $x_j - \frac{m_j}{M} \leq -\frac{1}{M}$ for some $j \in \{1, 2, \ldots, d\}$,*

$$|\phi_{\boldsymbol{m}}^{\tau}(\boldsymbol{x})| \leq (2\|\rho'\|_{\infty})^{d-1} \cdot 6\tau e^{-\tau}. \tag{33}$$

*Proof of Lemma A.6.* By Lagrange's Mean Value Theorem, there exist $\xi_i, \zeta_i \in [1, 2]$ $(i = 1, 2, \ldots, d)$ such that

$$
\begin{aligned}
|\phi_{\boldsymbol{m}}^{\tau}(\boldsymbol{x})| &= \prod_{i=1}^{d} \left| \psi^{\tau}\left(3M\left(x_i - \frac{m_i}{M}\right)\right)\right| \\
&= \prod_{i=1}^{d} \left| \rho'\left(3M\tau\left(x_i - \frac{m_i}{M}\right) + \xi_i\tau\right) - \rho'\left(3M\tau\left(x_i - \frac{m_i}{M}\right) - \zeta_i\tau\right)\right| \\
&\leq (2\|\rho'\|_{\infty})^{d-1} \left| \rho'\left(3M\tau\left(x_j - \frac{m_j}{M}\right) + \xi_j\tau\right) - \rho'\left(3M\tau\left(x_j - \frac{m_j}{M}\right) - \zeta_j\tau\right)\right| \\
&\leq (2\|\rho'\|_{\infty})^{d-1} \left( \left| \rho'\left(3M\tau\left(x_j - \frac{m_j}{M}\right) + \xi_j\tau\right) - 1\right| + \left| 1 - \rho'\left(3M\tau\left(x_j - \frac{m_j}{M}\right) - \zeta_j\tau\right)\right|\right).
\end{aligned}
\tag{34}
$$

Because $x_j - \frac{m_j}{M} \leq -\frac{1}{M}$ and $\tau \geq 1$, we have

$$3M\tau\left(x_j - \frac{m_j}{M}\right) - \zeta_j\tau \leq 3M\tau\left(x_j - \frac{m_j}{M}\right) + \xi_j\tau \leq -3\tau + \xi_j\tau \leq -\tau \leq -1. \tag{35}$$

Together with the fact that $-xe^x$ increases monotonically on $(-\infty, -1]$, it follows

$$
\begin{aligned}
|\phi_{\boldsymbol{m}}^{\tau}(\boldsymbol{x})| &\leq (2\|\rho'\|_{\infty})^{d-1} \left( \left| \rho'\left(3M\tau\left(x_j - \frac{m_j}{M}\right) + \xi_j\tau\right) - 1\right| + \left| 1 - \rho'\left(3M\tau\left(x_j - \frac{m_j}{M}\right) - \zeta_j\tau\right)\right|\right) \\
&\leq (2\|\rho'\|_{\infty})^{d-1} \left( -3\left(3M\tau\left(x_j - \frac{m_j}{M}\right) + \xi_j\tau\right) e^{3M\tau\left(x_j - \frac{m_j}{M}\right) + \xi_j\tau} + \right. \\
&\quad \left. -3\left(3M\tau\left(x_j - \frac{m_j}{M}\right) - \zeta_j\tau\right) e^{3M\tau\left(x_j - \frac{m_j}{M}\right) - \zeta_j\tau}\right) \qquad \text{(by Lemma A.3)} \\
&\leq (2\|\rho'\|_{\infty})^{d-1} \cdot 6\tau e^{-\tau}.
\end{aligned}
\tag{36}
$$

$\square$

Then the following Lemma A.7 follows directly from Lemma A.5 and Lemma A.6.

**Lemma A.7** (Locality of $\phi_{\boldsymbol{m}}^{\tau}$)**.** *Let $\tau \geq 1$. If $\left|x_j - \frac{m_j}{M}\right| \geq \frac{1}{M}$ for some $j \in \{1, 2, \ldots, d\}$,*

$$|\phi_{\boldsymbol{m}}^{\tau}(\boldsymbol{x})| \leq (2\|\rho'\|_{\infty})^{d-1} \cdot 6\tau e^{-\tau}. \tag{37}$$

**Lemma A.8** (Partition of unity property of $\phi_{\boldsymbol{m}}^{\tau}$)**.** *For all $\boldsymbol{x} \in [0, 1]^d$,*

$$\left| 1 - \sum_{\boldsymbol{m} \in [M]^d} \phi_{\boldsymbol{m}}^{\tau}(\boldsymbol{x})\right| \leq 6\tau e^{-\tau} \cdot \frac{(2\|\rho'\|_{\infty})^d - 1}{2\|\rho'\|_{\infty} - 1}. \tag{38}$$

*Proof of Lemma A.8.* Let $\tau \geq 1$. For all $\boldsymbol{x} \in [0, 1]^d$, by Lagrange's mean value theorem, there exist $\xi_i, \zeta_i \in [1, 2]$

$(i = 1, 2, \ldots, d)$ such that

$$
\left| 1 - \sum_{\boldsymbol{m} \in [M]^d} \phi_{\boldsymbol{m}}^{\tau}(\boldsymbol{x}) \right|
$$

$$
= \left| 1 - \sum_{\boldsymbol{m} \in [M]^d} \prod_{i=1}^{d} \psi^{\tau} \left( 3M \left( x_i - \frac{m_i}{M} \right) \right) \right|
$$

$$
= \left| 1 - \prod_{i=1}^{d} \sum_{m_i=0}^{M} \psi^{\tau} \left( 3M \left( x_i - \frac{m_i}{M} \right) \right) \right|
$$

$$
= \left| 1 - \prod_{i=1}^{d} \sum_{m_i=0}^{M} \frac{\rho(3M\tau(x_i - \frac{m_i}{M}) + 2\tau) - \rho(3M\tau(x_i - \frac{m_i}{M}) + \tau) - \rho(3M\tau(x_i - \frac{m_i}{M}) - \tau) + \rho(3M\tau(x_i - \frac{m_i}{M}) - 2\tau)}{\tau} \right|
$$

$$
\leq \left| 1 - \prod_{i=1}^{d} \frac{\rho(3M\tau x_i + 2\tau) - \rho(3M\tau x_i + \tau) - \rho(3M\tau(x_i - 1) - \tau) + \rho(3M\tau(x_i - 1) - 2\tau)}{\tau} \right|
$$

$$
= \left| 1 - \prod_{i=1}^{d} (\rho'(3M\tau x_i + \xi_i \tau) - \rho'(3M\tau(x_i - 1) - \zeta_i \tau)) \right|.
$$

By the inequality that

$$
\left| 1 - \prod_{i=1}^{d} x_i \right| = \left| 1 - x_1 + x_1 - x_1 x_2 + \cdots + \prod_{i=1}^{d-1} x_i - \prod_{i=1}^{d} x_i \right|
$$

$$
\leq |1 - x_1| + |x_1| \cdot |1 - x_2| + \cdots + \left| \prod_{i=1}^{d-1} x_i \right| \cdot |1 - x_d| \tag{39}
$$

and the fact that $3M\tau x_i + \xi_i \tau \geq \tau \geq 1$ and $3M\tau(x_i - 1) - \zeta_i \tau \leq -\tau \leq -1$ $(i = 1, 2, \ldots, d)$, it follows that

$$
\left| 1 - \sum_{\boldsymbol{m} \in [M]^d} \phi_{\boldsymbol{m}}^{\tau}(\boldsymbol{x}) \right|
$$

$$
\leq |1 - \rho'(3M\tau x_1 + \xi_1 \tau) + \rho'(3M\tau(x_1 - 1) - \zeta_1 \tau)| +
$$

$$
|\rho'(3M\tau x_1 + \xi_1 \tau) - \rho'(3M\tau(x_1 - 1) - \zeta_1 \tau)| \cdot |1 - \rho'(3M\tau x_2 + \xi_2 \tau) + \rho'(3M\tau(x_2 - 1) - \zeta_2 \tau)| + \cdots +
$$

$$
\prod_{i=1}^{d-1} |\rho'(3M\tau x_i + \xi_i \tau) - \rho'(3M\tau(x_i - 1) - \zeta_i \tau)| \cdot |1 - \rho'(3M\tau x_d + \xi_d \tau) + \rho'(3M\tau(x_d - 1) - \zeta_d \tau)|
$$

$$
\leq 6\tau e^{-\tau} \cdot (1 + 2\|\rho'\|_{\infty} + \cdots + (2\|\rho'\|_{\infty})^{d-1}) \qquad \text{(by Lemma A.2 and A.3)}
$$

$$
= 6\tau e^{-\tau} \cdot \frac{(2\|\rho'\|_{\infty})^d - 1}{2\|\rho'\|_{\infty} - 1}.
$$

$\square$

At the end of this section, we prove Lemma 5.2, that is, $\sum_{\boldsymbol{m} \in [M]^d} P_{\boldsymbol{m}}^{\kappa} \phi_{\boldsymbol{m}}^{\tau}$ approximates $f \in \mathcal{C}^{\beta,R}([0,1]^d)$.

*Proof of Lemma 5.2.* For all $\boldsymbol{x} \in [0,1]^d$,

$$\left| f(\boldsymbol{x}) - \sum_{\boldsymbol{m} \in [M]^d} P_{\boldsymbol{m}}^\kappa(\boldsymbol{x}) \phi_{\boldsymbol{m}}^\tau(\boldsymbol{x}) \right|$$

$$\leq \left| f(\boldsymbol{x}) - \sum_{\boldsymbol{m} \in [M]^d} f(\boldsymbol{x}) \phi_{\boldsymbol{m}}^\tau(\boldsymbol{x}) \right| + \left| \sum_{\boldsymbol{m} \in [M]^d} f(\boldsymbol{x}) \phi_{\boldsymbol{m}}^\tau(\boldsymbol{x}) - \sum_{\boldsymbol{m} \in [M]^d} P_{\boldsymbol{m}}^\kappa(\boldsymbol{x}) \phi_{\boldsymbol{m}}^\tau(\boldsymbol{x}) \right|$$

$$\leq 6\tau e^{-\tau} \frac{(2\|\rho'\|_\infty)^d - 1}{2\|\rho'\|_\infty - 1} |f(\boldsymbol{x})| + \left| \sum_{\boldsymbol{m} \in [M]^d} \left( f(\boldsymbol{x}) - P_{\boldsymbol{m}}^\kappa(\boldsymbol{x}) \right) \phi_{\boldsymbol{m}}^\tau(\boldsymbol{x}) \right| \qquad \text{(by Lemma A.8)}$$

$$\leq 6\tau e^{-\tau} \frac{(2\|\rho'\|_\infty)^d - 1}{2\|\rho'\|_\infty - 1} R + \left| \sum_{\boldsymbol{m} \in [M]^d} \left( f(\boldsymbol{x}) - P_{\boldsymbol{m}}^\kappa(\boldsymbol{x}) \right) \phi_{\boldsymbol{m}}^\tau(\boldsymbol{x}) \right|.$$

For the second term,

$$\left| \sum_{\boldsymbol{m} \in [M]^d} \left( f(\boldsymbol{x}) - P_{\boldsymbol{m}}^\kappa(\boldsymbol{x}) \right) \phi_{\boldsymbol{m}}^\tau(\boldsymbol{x}) \right|$$

$$\leq \left| \sum_{\substack{\boldsymbol{m} \in [M]^d \\ \|\boldsymbol{x} - \frac{\boldsymbol{m}}{M}\|_\infty \leq \frac{1}{M}}} \left( f(\boldsymbol{x}) - P_{\boldsymbol{m}}^\kappa(\boldsymbol{x}) \right) \phi_{\boldsymbol{m}}^\tau(\boldsymbol{x}) \right| + \left| \sum_{\substack{\boldsymbol{m} \in [M]^d \\ \|\boldsymbol{x} - \frac{\boldsymbol{m}}{M}\|_\infty > \frac{1}{M}}} \left( f(\boldsymbol{x}) - P_{\boldsymbol{m}}^\kappa(\boldsymbol{x}) \right) \phi_{\boldsymbol{m}}^\tau(\boldsymbol{x}) \right|$$

$$\leq \binom{\kappa + d - 1}{d - 1} R M^{-\beta} \left| \sum_{\substack{\boldsymbol{m} \in [M]^d \\ \|\boldsymbol{x} - \frac{\boldsymbol{m}}{M}\|_\infty \leq \frac{1}{M}}} \phi_{\boldsymbol{m}}^\tau(\boldsymbol{x}) \right| + \binom{\kappa + d - 1}{d - 1} R \left| \sum_{\substack{\boldsymbol{m} \in [M]^d \\ \|\boldsymbol{x} - \frac{\boldsymbol{m}}{M}\|_\infty > \frac{1}{M}}} \phi_{\boldsymbol{m}}^\tau(\boldsymbol{x}) \right| \qquad \text{(by Lemma 5.1)}$$

$$\leq \binom{\kappa + d - 1}{d - 1} R M^{-\beta} 3^d (2\|\rho'\|_\infty)^d + \binom{\kappa + d - 1}{d - 1} R \left( (M+1)^d - 2^d \right) (2\|\rho'\|_\infty)^{d-1} 6\tau e^{-\tau} \qquad \text{(by Lemma A.7)}$$

$$\leq 3^d M^{-\beta} \binom{\kappa + d - 1}{d - 1} R (2\|\rho'\|_\infty)^d + 6(M+1)^d \tau e^{-\tau} \binom{\kappa + d - 1}{d - 1} R (2\|\rho'\|_\infty)^{d-1}.$$

$\square$

## A.2. Approximating $(P_{\boldsymbol{m}}^\kappa)_{\boldsymbol{m} \in [M]^d}$ by a Low-Rank Swish Network $\mathcal{P}$

We first prove Lemma 5.4 which shows how to approximate the square function with a Swish network of depth 1 and width 2 and Lemma 5.5 which shows how to approximate the multiplication function with a Swish network of depth 1 and width 4.

*Proof of Lemma 5.4.* For all $x \in \mathbb{R}$, by Taylor expansion theorem, there exist $\xi$ between 0 and $\frac{x}{\lambda}$ and $\zeta$ between 0 and $-\frac{x}{\lambda}$ such that

$$\rho\left(\frac{x}{\lambda}\right) = \rho(0) + \rho'(0) \cdot \frac{x}{\lambda} + \frac{\rho''(0)}{2!} \cdot \frac{x^2}{\lambda^2} + \frac{\rho'''(0)}{3!} \cdot \frac{x^3}{\lambda^3} + \frac{\rho^{(4)}(\xi)}{4!} \cdot \frac{x^4}{\lambda^4} \tag{40}$$

and

$$\rho\left(-\frac{x}{\lambda}\right) = \rho(0) - \rho'(0) \cdot \frac{x}{\lambda} + \frac{\rho''(0)}{2!} \cdot \frac{x^2}{\lambda^2} - \frac{\rho'''(0)}{3!} \cdot \frac{x^3}{\lambda^3} + \frac{\rho^{(4)}(\zeta)}{4!} \cdot \frac{x^4}{\lambda^4}. \tag{41}$$

This, together with $\rho(0) = 0$ and $\rho''(0) = \frac{1}{2}$, implies that

$$2\lambda^2 \left( \rho\left(\frac{x}{\lambda}\right) + \rho\left(-\frac{x}{\lambda}\right) \right) = x^2 + \frac{\rho^{(4)}(\xi) + \rho^{(4)}(\zeta)}{12} \cdot \frac{x^4}{\lambda^2}. \tag{42}$$

Then, by the fact that $\|\rho^{(4)}\|_\infty \leq \frac{1}{2}$, it follows

$$\left| 2\lambda^2 \left( \rho\left(\frac{x}{\lambda}\right) + \rho\left(-\frac{x}{\lambda}\right) \right) - x^2 \right| \leq \frac{2\|\rho^{(4)}\|_\infty x^4}{12\lambda^2} \leq \frac{x^4}{12\lambda^2}. \tag{43}$$

$\square$

*Proof of Lemma 5.5.* By Lemma 5.4, there exist $\xi_1$ between $0$ and $\frac{x+y}{2\lambda}$, $\zeta_1$ between $0$ and $-\frac{x+y}{2\lambda}$, $\xi_2$ between $0$ and $\frac{x-y}{2\lambda}$, and $\zeta_2$ between $0$ and $-\frac{x-y}{2\lambda}$ such that

$$\begin{aligned}
&\left| 2\lambda^2 \left( \rho\left(\frac{x+y}{2\lambda}\right) + \rho\left(-\frac{x+y}{2\lambda}\right) - \rho\left(\frac{x-y}{2\lambda}\right) - \rho\left(-\frac{x-y}{2\lambda}\right) \right) - xy \right| \\
&= \left| \frac{\rho^{(4)}(\xi_1) + \rho^{(4)}(\zeta_1)}{12\lambda^2} \left(\frac{x+y}{2}\right)^4 - \frac{\rho^{(4)}(\xi_2) + \rho^{(4)}(\zeta_2)}{12\lambda^2} \left(\frac{x-y}{2}\right)^4 \right| \\
&\leq \frac{2\|\rho^{(4)}\|_\infty}{12\lambda^2} \left(\frac{x+y}{2}\right)^4 + \frac{2\|\rho^{(4)}\|_\infty}{12\lambda^2} \left(\frac{x-y}{2}\right)^4 \\
&\leq \frac{1}{12\lambda^2} \left( \left(\frac{x+y}{2}\right)^4 + \left(\frac{x-y}{2}\right)^4 \right) \\
&= \frac{1}{12\lambda^2} \cdot \frac{x^4 + 6x^2y^2 + y^4}{8}.
\end{aligned} \tag{44}$$

$\square$

For convenience, we denote

$$\begin{aligned}
id(x) &:= \rho(x) - \rho(-x), \\
sq(x) &:= 2\lambda^2 \left( \rho\left(\frac{x}{\lambda}\right) + \rho\left(-\frac{x}{\lambda}\right) \right),
\end{aligned}$$

and

$$mult(x) := sq\left(\frac{x+y}{2}\right) - sq\left(\frac{x-y}{2}\right) = 2\lambda^2 \left( \rho\left(\frac{x+y}{2\lambda}\right) + \rho\left(-\frac{x+y}{2\lambda}\right) - \rho\left(\frac{x-y}{2\lambda}\right) - \rho\left(-\frac{x-y}{2\lambda}\right) \right).$$

Obviously, $id$ can be implemented by a network of depth 1, width 2, number of nonzero parameters 4, and maximum absolute value of parameters 1, $sq$ can be implemented by a network of depth 1, width 2, number of nonzero parameters 4, and maximum absolute value of parameters $\max\{2\lambda^2, \frac{1}{\lambda}\}$, and $mult$ can be implemented by a network of depth 1, width 4, number of nonzero parameters 12, and maximum absolute value of parameters $\max\{2\lambda^2, \frac{1}{2\lambda}\}$.

Next we want to construct monomials by stacking $id$, $sq$, and $mult$. To prepare for the subsequent approximation error analysis, we show some conclusions about the output ranges of $sq$ and $mult$.

**Lemma A.9** (The output range of $sq$). *Let $\lambda > 0$. For all $x \in \mathbb{R}$,*

$$0 \leq sq(x) \leq x^2. \tag{45}$$

*Proof of Lemma A.9.* (1). We first prove $0 \leq sq(x)$ for all $x \in \mathbb{R}$. The derivative of $sq$ is

$$\begin{aligned}
sq'(x) &= 2\lambda \left( \rho'\left(\frac{x}{\lambda}\right) - \rho'\left(-\frac{x}{\lambda}\right) \right) \\
&= 2\lambda \left( \frac{1 + e^{-\frac{x}{\lambda}} + \frac{x}{\lambda}e^{-\frac{x}{\lambda}}}{(1 + e^{-\frac{x}{\lambda}})^2} - \frac{1 + e^{\frac{x}{\lambda}} - \frac{x}{\lambda}e^{\frac{x}{\lambda}}}{(1 + e^{\frac{x}{\lambda}})^2} \right) \\
&= 2\lambda \cdot \frac{e^{\frac{2x}{\lambda}} + \frac{2x}{\lambda}e^{\frac{x}{\lambda}} - 1}{(1 + e^{\frac{x}{\lambda}})^2}.
\end{aligned}$$

Let $g(x) := e^{\frac{2x}{\lambda}} + \frac{2x}{\lambda}e^{\frac{x}{\lambda}} - 1$. For all $x \in \mathbb{R}$, the sign of $sq'(x)$ is consistent with that of $g(x)$, because $\frac{2\lambda}{(1+e^{\frac{x}{\lambda}})} > 0$. When $x < 0$, $g(x) < e^{\frac{2x}{\lambda}} - 1 < e^0 - 1 = 0$. ; when $x > 0$, $g(x) > e^{\frac{2x}{\lambda}} - 1 > e^0 - 1 = 0$. Therefore, $sq$ is monotonically decreasing on $(-\infty, 0)$ and increasing on $(0, +\infty)$. It follows that for all $x \in \mathbb{R}$, $sq(x) \geq sq(0) = 0$.

(2). Next we prove $sq(x) \leq x^2$ for all $x \in \mathbb{R}$. The derivative of $x^2 - sq(x)$ is

$$\frac{d}{dx}(x^2 - sq(x)) = 2x - 2\lambda^2 \cdot \frac{e^{\frac{2x}{\lambda}} + \frac{2x}{\lambda}e^{\frac{x}{\lambda}} - 1}{(1 + e^{\frac{x}{\lambda}})^2}$$
$$= \frac{2x + 2xe^{\frac{2x}{\lambda}} - 2\lambda e^{\frac{2x}{\lambda}} + 2\lambda}{(1 + e^{\frac{x}{\lambda}})^2}.$$

Let $h(x) := 2x + 2xe^{\frac{2x}{\lambda}} - 2\lambda e^{\frac{2x}{\lambda}} + 2\lambda$. Then the derivative of $h$ is

$$h'(x) = 2 + 2e^{\frac{2x}{\lambda}} + \frac{4x}{\lambda}e^{\frac{2x}{\lambda}} - 2e^{\frac{2x}{\lambda}}$$

and the 2nd derivative is

$$h''(x) = \frac{8x}{\lambda^2}e^{\frac{2x}{\lambda}}.$$

When $x < 0$, $h''(x) < 0$; when $x > 0$, $h''(x) > 0$. Therefore, $h'$ is monotonically decreasing on $(-\infty, 0)$ and increasing on $(0, +\infty)$. Then for all $x \in \mathbb{R}$, $h'(x) \geq h'(0) = 0$. It follows that $h$ is monotonically increasing on $\mathbb{R}$. By the fact that $h(0) = 0$, we have $h(x) \leq 0$ when $x < 0$ and $h(x) \geq 0$ when $x > 0$. It implies that $x^2 - sq(x)$ is monotonically decreasing on $(-\infty, 0)$ and increasing on $(0, +\infty)$. Finally, we have $x^2 - sq(x) \geq 0 - sq(0) = 0$ for all $x \in \mathbb{R}$. $\square$

**Lemma A.10** (The output range of $mult$. part I). *Let $\lambda > 0$. For all $x, y \geq 0$, $0 \leq mult(x, y) \leq xy$.*

*Proof of Lemma A.10.* From the proof of Lemma A.9, we know that both $sq(x)$ and $x^2 - sq(x)$ are monotonically increasing on $(0, +\infty)$. Therefore, for all $x, y \geq 0$, when $x - y \geq 0$, since $x + y \geq x - y \geq 0$,

$$sq\left(\frac{x+y}{2}\right) \geq sq\left(\frac{x-y}{2}\right) \Rightarrow mult(x, y) = sq\left(\frac{x+y}{2}\right) - sq\left(\frac{x-y}{2}\right) \geq 0,$$
$$\left(\frac{x+y}{2}\right)^2 - sq\left(\frac{x+y}{2}\right) \geq \left(\frac{x-y}{2}\right)^2 - sq\left(\frac{x-y}{2}\right) \Rightarrow$$
$$sq\left(\frac{x+y}{2}\right) - sq\left(\frac{x-y}{2}\right) \leq \left(\frac{x+y}{2}\right)^2 - \left(\frac{x-y}{2}\right)^2 = xy \Rightarrow mult(x, y) \leq xy,$$

and when $x - y \leq 0$, since $x + y \geq y - x \geq 0$,

$$sq\left(\frac{x+y}{2}\right) - sq\left(\frac{x-y}{2}\right) = sq\left(\frac{x+y}{2}\right) - sq\left(\frac{y-x}{2}\right) \geq 0 \Rightarrow mult(x, y) \geq 0,$$
$$\left(\frac{x+y}{2}\right)^2 - sq\left(\frac{x+y}{2}\right) \geq \left(\frac{y-x}{2}\right)^2 - sq\left(\frac{y-x}{2}\right) = \left(\frac{x-y}{2}\right)^2 - sq\left(\frac{x-y}{2}\right) \Rightarrow$$
$$sq\left(\frac{x+y}{2}\right) - sq\left(\frac{x-y}{2}\right) \leq \left(\frac{x+y}{2}\right)^2 - \left(\frac{x-y}{2}\right)^2 = xy \Rightarrow mult(x, y) \leq xy.$$

$\square$

Similar to the proof of Lemma A.10, it is easy to prove the following three lemmas.

**Lemma A.11** (The output range of $mult$. part II). *Let $\lambda > 0$. For all $x \leq 0$ and $y \geq 0$, $xy \leq mult(x, y) \leq 0$.*

**Lemma A.12** (The output range of $mult$. part III). *Let $\lambda > 0$. For all $x, y \leq 0$, $0 \leq mult(x, y) \leq xy$.*

**Lemma A.13** (The output range of $mult$. part IV). *Let $\lambda > 0$. For all $x \geq 0$ and $y \leq 0$, $xy \leq mult(x, y) \leq 0$.*

Combining Lemma A.10, A.11, A.12, and A.13, we have:

**Lemma A.14** (The output range of $mult$). *Let $\lambda > 0$. For all $x, y \in \mathbb{R}$, $|mult(x, y)| \leq |xy|$.*

Next we define a series of functions $\mathcal{M}_{\boldsymbol{\alpha}}$ where $\boldsymbol{\alpha} \in \mathbb{N}_+^d$ by stacking $sq$ and $mult$. These functions will be implemented by hidden neurons of $\mathcal{P}$ that is a network outputting Taylor polynomials at all grid points. For any $\boldsymbol{\alpha} \in \mathbb{N}_+^d$, the function $\mathcal{M}_{\boldsymbol{\alpha}}$ is defined by:

1. when $|\boldsymbol{\alpha}| = 1$, $\mathcal{M}(\boldsymbol{x}) := \boldsymbol{x}^{\boldsymbol{\alpha}}$;

2. when $|\boldsymbol{\alpha}| = 2$ and $\exists |\boldsymbol{\alpha}'| = 1$ such that $\boldsymbol{\alpha} = 2\boldsymbol{\alpha}'$, $\mathcal{M}(\boldsymbol{x}) := sq(\mathcal{M}_{\boldsymbol{\alpha}'}(\boldsymbol{x}))$;

3. when $|\boldsymbol{\alpha}| = 2$ and $\exists |\boldsymbol{\alpha}'| = |\boldsymbol{\alpha}''| = 1$ such that $\boldsymbol{\alpha} = \boldsymbol{\alpha}' + \boldsymbol{\alpha}''$ and $\boldsymbol{\alpha}' \neq \boldsymbol{\alpha}''$, $\mathcal{M}(\boldsymbol{x}) := mult(\mathcal{M}_{\boldsymbol{\alpha}'}(\boldsymbol{x}), \mathcal{M}_{\boldsymbol{\alpha}''}(\boldsymbol{x}))$;

4. when $|\boldsymbol{\alpha}| = 4$ and $\exists |\boldsymbol{\alpha}'| = 2$ such that $\boldsymbol{\alpha} = 2\boldsymbol{\alpha}'$, $\mathcal{M}(\boldsymbol{x}) := sq(\mathcal{M}_{\boldsymbol{\alpha}'}(\boldsymbol{x}))$;

5. when $|\boldsymbol{\alpha}| = 4$ and $\exists |\boldsymbol{\alpha}'| = |\boldsymbol{\alpha}''| = 2$ such that $\boldsymbol{\alpha} = \boldsymbol{\alpha}' + \boldsymbol{\alpha}''$ and $\boldsymbol{\alpha}' \neq \boldsymbol{\alpha}''$, $\mathcal{M}(\boldsymbol{x}) := mult(\mathcal{M}_{\boldsymbol{\alpha}'}(\boldsymbol{x}), \mathcal{M}_{\boldsymbol{\alpha}''}(\boldsymbol{x}))$;

6. when $|\boldsymbol{\alpha}| \geq 5$ or $= 3$ and $\exists |\boldsymbol{\alpha}'| = 2, |\boldsymbol{\alpha}''| = |\boldsymbol{\alpha}| - |\boldsymbol{\alpha}''|$ such that $\boldsymbol{\alpha} = \boldsymbol{\alpha}' + \boldsymbol{\alpha}''$, $\mathcal{M}(\boldsymbol{x}) := mult(\mathcal{M}_{\boldsymbol{\alpha}'}(\boldsymbol{x}), \mathcal{M}_{\boldsymbol{\alpha}''}(\boldsymbol{x}))$.

Next we show the upper bound of $\mathcal{M}_{\boldsymbol{\alpha}}$ on $[-1, 1]^d$ using Lemma A.9 and A.14.

**Lemma A.15** (The upper bound of $|\mathcal{M}_{\boldsymbol{\alpha}}|$). *Let $\lambda > 0$ and $\boldsymbol{\alpha} \in \mathbb{N}^d$. For all $\boldsymbol{x} \in [-1, 1]^d$,*

$$|\mathcal{M}_{\boldsymbol{\alpha}}(\boldsymbol{x})| \leq 1. \tag{46}$$

*Proof of Lemma A.15.* Here we prove it by mathematical induction. When $|\boldsymbol{\alpha}| = 1$,

$$|\mathcal{M}_{\boldsymbol{\alpha}}(\boldsymbol{x})| = |\boldsymbol{x}^{\boldsymbol{\alpha}}| \leq 1.$$

When $|\boldsymbol{\alpha}| = 2$ and $\exists |\boldsymbol{\alpha}'| = 1$ such that $\boldsymbol{\alpha} = 2\boldsymbol{\alpha}'$, by Lemma A.9,

$$|\mathcal{M}_{\boldsymbol{\alpha}}(\boldsymbol{x})| = |sq(\mathcal{M}_{\boldsymbol{\alpha}'}(\boldsymbol{x}))| \leq (\mathcal{M}_{\boldsymbol{\alpha}'}(\boldsymbol{x}))^2 \leq 1.$$

When $|\boldsymbol{\alpha}| = 2$ and $\exists |\boldsymbol{\alpha}'| = |\boldsymbol{\alpha}''| = 1$ such that $\boldsymbol{\alpha} = \boldsymbol{\alpha}' + \boldsymbol{\alpha}''$ and $\boldsymbol{\alpha}' \neq \boldsymbol{\alpha}''$, by Lemma A.14,

$$|\mathcal{M}_{\boldsymbol{\alpha}}(\boldsymbol{x})| = |mult(\mathcal{M}_{\boldsymbol{\alpha}'}(\boldsymbol{x}), \mathcal{M}_{\boldsymbol{\alpha}''}(\boldsymbol{x}))| \leq |\mathcal{M}_{\boldsymbol{\alpha}'}(\boldsymbol{x}) \cdot \mathcal{M}_{\boldsymbol{\alpha}''}(\boldsymbol{x})| \leq 1.$$

When $|\boldsymbol{\alpha}| = 3$ and $\exists |\boldsymbol{\alpha}'| = 2, |\boldsymbol{\alpha}''| = |\boldsymbol{\alpha}| - |\boldsymbol{\alpha}''|$ such that $\boldsymbol{\alpha} = \boldsymbol{\alpha}' + \boldsymbol{\alpha}''$, by Lemma A.14,

$$|\mathcal{M}_{\boldsymbol{\alpha}}(\boldsymbol{x})| = |mult(\mathcal{M}_{\boldsymbol{\alpha}'}(\boldsymbol{x}), \mathcal{M}_{\boldsymbol{\alpha}''}(\boldsymbol{x}))| \leq |\mathcal{M}_{\boldsymbol{\alpha}'}(\boldsymbol{x}) \cdot \mathcal{M}_{\boldsymbol{\alpha}''}(\boldsymbol{x})| \leq 1.$$

When $|\boldsymbol{\alpha}| = 4$ and $\exists |\boldsymbol{\alpha}'| = 2$ such that $\boldsymbol{\alpha} = 2\boldsymbol{\alpha}'$, by Lemma A.14,

$$|\mathcal{M}_{\boldsymbol{\alpha}}(\boldsymbol{x})| = |sq(\mathcal{M}_{\boldsymbol{\alpha}'}(\boldsymbol{x}))| \leq (\mathcal{M}_{\boldsymbol{\alpha}'}(\boldsymbol{x}))^2 \leq 1.$$

When $|\boldsymbol{\alpha}| = 4$ and $\exists |\boldsymbol{\alpha}'| = |\boldsymbol{\alpha}''| = 2$ such that $\boldsymbol{\alpha} = \boldsymbol{\alpha}' + \boldsymbol{\alpha}''$ and $\boldsymbol{\alpha}' \neq \boldsymbol{\alpha}''$, by Lemma A.14,

$$|\mathcal{M}_{\boldsymbol{\alpha}}(\boldsymbol{x})| = |mult(\mathcal{M}_{\boldsymbol{\alpha}'}(\boldsymbol{x}), \mathcal{M}_{\boldsymbol{\alpha}''}(\boldsymbol{x}))| \leq |\mathcal{M}_{\boldsymbol{\alpha}'}(\boldsymbol{x}) \cdot \mathcal{M}_{\boldsymbol{\alpha}''}(\boldsymbol{x})| \leq 1.$$

When $|\boldsymbol{\alpha}| \geq 5$ and $\exists |\boldsymbol{\alpha}'| = 2, |\boldsymbol{\alpha}''| = |\boldsymbol{\alpha}| - |\boldsymbol{\alpha}''|$ such that $\boldsymbol{\alpha} = \boldsymbol{\alpha}' + \boldsymbol{\alpha}''$, by Lemma A.14 and induction,

$$|\mathcal{M}_{\boldsymbol{\alpha}}(\boldsymbol{x})| = |mult(\mathcal{M}_{\boldsymbol{\alpha}'}(\boldsymbol{x}), \mathcal{M}_{\boldsymbol{\alpha}''}(\boldsymbol{x}))| \leq |\mathcal{M}_{\boldsymbol{\alpha}'}(\boldsymbol{x}) \cdot \mathcal{M}_{\boldsymbol{\alpha}''}(\boldsymbol{x})| \leq 1.$$

$\square$

The following lemma shows the error of $\mathcal{M}_{\boldsymbol{\alpha}}(\boldsymbol{x})$ to approximate $\boldsymbol{x}^{\boldsymbol{\alpha}}$ measured by sup norm on $[-1, 1]^d$.

**Lemma A.16.** *Let $\lambda > 0$ and $\boldsymbol{\alpha} \in \mathbb{N}^d$. Then for all $\boldsymbol{x} \in [-1, 1]^d$,*

$$|\mathcal{M}_{\boldsymbol{\alpha}}(\boldsymbol{x}) - \boldsymbol{x}^{\boldsymbol{\alpha}}| \leq \frac{|\boldsymbol{\alpha}| - 1}{12\lambda^2}. \tag{47}$$

*Proof of Lemma A.16.* Here we prove it by mathematical induction. When $|\boldsymbol{\alpha}| = 1$,

$$|\mathcal{M}_{\boldsymbol{\alpha}} - \boldsymbol{x}^{\boldsymbol{\alpha}}| = |\boldsymbol{x}^{\boldsymbol{\alpha}} - \boldsymbol{x}^{\boldsymbol{\alpha}}| = 0.$$

When $|\boldsymbol{\alpha}| = 2$ and $\exists |\boldsymbol{\alpha}'| = 1$ such that $\boldsymbol{\alpha} = 2\boldsymbol{\alpha}'$, by Lemma 5.4,

$$|\mathcal{M}_{\boldsymbol{\alpha}}(\boldsymbol{x}) - \boldsymbol{x}^{\boldsymbol{\alpha}}| = |sq(\mathcal{M}_{\boldsymbol{\alpha}'}(\boldsymbol{x})) - \boldsymbol{x}^{2\boldsymbol{\alpha}'}| = |sq(\boldsymbol{x}^{\boldsymbol{\alpha}'}) - \boldsymbol{x}^{2\boldsymbol{\alpha}'}| \leq \frac{\boldsymbol{x}^{4\boldsymbol{\alpha}'}}{12\lambda^2} \leq \frac{1}{12\lambda^2}.$$

When $|\boldsymbol{\alpha}| = 2$ and $\exists |\boldsymbol{\alpha}'| = |\boldsymbol{\alpha}''| = 1$ such that $\boldsymbol{\alpha} = \boldsymbol{\alpha}' + \boldsymbol{\alpha}''$ and $\boldsymbol{\alpha}' \neq \boldsymbol{\alpha}''$, by Lemma 5.5,

$$\begin{aligned}
|\mathcal{M}_{\boldsymbol{\alpha}}(\boldsymbol{x}) - \boldsymbol{x}^{\boldsymbol{\alpha}}| &= |mult(\mathcal{M}_{\boldsymbol{\alpha}'}(\boldsymbol{x}), \mathcal{M}_{\boldsymbol{\alpha}''}(\boldsymbol{x})) - \boldsymbol{x}^{\boldsymbol{\alpha}'} \cdot \boldsymbol{x}^{\boldsymbol{\alpha}''}| \\
&= |mult(\boldsymbol{x}^{\boldsymbol{\alpha}'}, \boldsymbol{x}^{\boldsymbol{\alpha}''}) - \boldsymbol{x}^{\boldsymbol{\alpha}'} \cdot \boldsymbol{x}^{\boldsymbol{\alpha}''}| \\
&\leq \frac{1}{12\lambda^2} \cdot \frac{\boldsymbol{x}^{4\boldsymbol{\alpha}'} + 6\boldsymbol{x}^{2\boldsymbol{\alpha}'} \cdot \boldsymbol{x}^{2\boldsymbol{\alpha}''} + \boldsymbol{x}^{4\boldsymbol{\alpha}''}}{8} \\
&\leq \frac{1}{12\lambda^2}.
\end{aligned}$$

When $|\boldsymbol{\alpha}| = 3$ and $\exists |\boldsymbol{\alpha}'| = 2, |\boldsymbol{\alpha}''| = |\boldsymbol{\alpha}| - |\boldsymbol{\alpha}''|$ such that $\boldsymbol{\alpha} = \boldsymbol{\alpha}' + \boldsymbol{\alpha}''$, by Lemma 5.5 and A.15,

$$\begin{aligned}
&|\mathcal{M}_{\boldsymbol{\alpha}}(\boldsymbol{x}) - \boldsymbol{x}^{\boldsymbol{\alpha}}| \\
&= |mult(\mathcal{M}_{\boldsymbol{\alpha}'}(\boldsymbol{x}), \mathcal{M}_{\boldsymbol{\alpha}''}(\boldsymbol{x})) - \boldsymbol{x}^{\boldsymbol{\alpha}'} \cdot \boldsymbol{x}^{\boldsymbol{\alpha}''}| \\
&\leq |mult(\mathcal{M}_{\boldsymbol{\alpha}'}(\boldsymbol{x}), \mathcal{M}_{\boldsymbol{\alpha}''}(\boldsymbol{x})) - \mathcal{M}_{\boldsymbol{\alpha}'}(\boldsymbol{x}) \cdot \mathcal{M}_{\boldsymbol{\alpha}''}(\boldsymbol{x})| + |\mathcal{M}_{\boldsymbol{\alpha}''}(\boldsymbol{x})| \cdot |\mathcal{M}_{\boldsymbol{\alpha}'}(\boldsymbol{x}) - \boldsymbol{x}^{\boldsymbol{\alpha}'}| + |\boldsymbol{x}^{\boldsymbol{\alpha}'}| \cdot |\mathcal{M}_{\boldsymbol{\alpha}''}(\boldsymbol{x}) - \boldsymbol{x}^{\boldsymbol{\alpha}''}| \\
&\leq \frac{1}{12\lambda^2} + \frac{1}{12\lambda^2} + 0 \\
&= \frac{2}{12\lambda^2}.
\end{aligned}$$

When $|\boldsymbol{\alpha}| = 4$ and $\exists |\boldsymbol{\alpha}'| = 2$ such that $\boldsymbol{\alpha} = 2\boldsymbol{\alpha}'$, by Lemma 5.4 and A.15,

$$\begin{aligned}
&|\mathcal{M}_{\boldsymbol{\alpha}}(\boldsymbol{x}) - \boldsymbol{x}^{\boldsymbol{\alpha}}| \\
&\leq |sq(\mathcal{M}_{\boldsymbol{\alpha}'}(\boldsymbol{x})) - (\mathcal{M}_{\boldsymbol{\alpha}'}(\boldsymbol{x}))^2| + |(\mathcal{M}_{\boldsymbol{\alpha}'}(\boldsymbol{x}))^2 - \boldsymbol{x}^{2\boldsymbol{\alpha}'}| \\
&\leq \frac{1}{12\lambda^2} + |\mathcal{M}_{\boldsymbol{\alpha}'}(\boldsymbol{x})| \cdot |\mathcal{M}_{\boldsymbol{\alpha}'}(\boldsymbol{x}) - \boldsymbol{x}^{\boldsymbol{\alpha}'}| + |\boldsymbol{x}^{\boldsymbol{\alpha}'}| \cdot |\mathcal{M}_{\boldsymbol{\alpha}'}(\boldsymbol{x}) - \boldsymbol{x}^{\boldsymbol{\alpha}'}| \\
&\leq \frac{3}{12\lambda^2}.
\end{aligned}$$

When $|\boldsymbol{\alpha}| = 4$ and $\exists |\boldsymbol{\alpha}'| = |\boldsymbol{\alpha}''| = 2$ such that $\boldsymbol{\alpha} = \boldsymbol{\alpha}' + \boldsymbol{\alpha}''$ and $\boldsymbol{\alpha}' \neq \boldsymbol{\alpha}''$, by Lemma 5.5 and A.15,

$$\begin{aligned}
&|\mathcal{M}_{\boldsymbol{\alpha}}(\boldsymbol{x}) - \boldsymbol{x}^{\boldsymbol{\alpha}}| \\
&= |mult(\mathcal{M}_{\boldsymbol{\alpha}'}(\boldsymbol{x}), \mathcal{M}_{\boldsymbol{\alpha}''}(\boldsymbol{x})) - \boldsymbol{x}^{\boldsymbol{\alpha}'} \cdot \boldsymbol{x}^{\boldsymbol{\alpha}''}| \\
&\leq |mult(\mathcal{M}_{\boldsymbol{\alpha}'}(\boldsymbol{x}), \mathcal{M}_{\boldsymbol{\alpha}''}(\boldsymbol{x})) - \mathcal{M}_{\boldsymbol{\alpha}'}(\boldsymbol{x}) \cdot \mathcal{M}_{\boldsymbol{\alpha}''}(\boldsymbol{x})| + |\mathcal{M}_{\boldsymbol{\alpha}''}(\boldsymbol{x})| \cdot |\mathcal{M}_{\boldsymbol{\alpha}'}(\boldsymbol{x}) - \boldsymbol{x}^{\boldsymbol{\alpha}'}| + |\boldsymbol{x}^{\boldsymbol{\alpha}'}| \cdot |\mathcal{M}_{\boldsymbol{\alpha}''}(\boldsymbol{x}) - \boldsymbol{x}^{\boldsymbol{\alpha}''}| \\
&\leq \frac{1}{12\lambda^2} + \frac{1}{12\lambda^2} + \frac{1}{12\lambda^2} \\
&= \frac{3}{12\lambda^2}.
\end{aligned}$$

When $|\boldsymbol{\alpha}| \geq 5$ and $\exists |\boldsymbol{\alpha}'| = 2, |\boldsymbol{\alpha}''| = |\boldsymbol{\alpha}| - |\boldsymbol{\alpha}''|$ such that $\boldsymbol{\alpha} = \boldsymbol{\alpha}' + \boldsymbol{\alpha}''$, by Lemma 5.5 and A.15 and induction,

$$\begin{aligned}
&|\mathcal{M}_{\boldsymbol{\alpha}}(\boldsymbol{x}) - \boldsymbol{x}^{\boldsymbol{\alpha}}| \\
&= |mult(\mathcal{M}_{\boldsymbol{\alpha}'}(\boldsymbol{x}), \mathcal{M}_{\boldsymbol{\alpha}''}(\boldsymbol{x})) - \boldsymbol{x}^{\boldsymbol{\alpha}'} \cdot \boldsymbol{x}^{\boldsymbol{\alpha}''}| \\
&\leq |mult(\mathcal{M}_{\boldsymbol{\alpha}'}(\boldsymbol{x}), \mathcal{M}_{\boldsymbol{\alpha}''}(\boldsymbol{x})) - \mathcal{M}_{\boldsymbol{\alpha}'}(\boldsymbol{x}) \cdot \mathcal{M}_{\boldsymbol{\alpha}''}(\boldsymbol{x})| + |\mathcal{M}_{\boldsymbol{\alpha}''}(\boldsymbol{x})| \cdot |\mathcal{M}_{\boldsymbol{\alpha}'}(\boldsymbol{x}) - \boldsymbol{x}^{\boldsymbol{\alpha}'}| + |\boldsymbol{x}^{\boldsymbol{\alpha}'}| \cdot |\mathcal{M}_{\boldsymbol{\alpha}''}(\boldsymbol{x}) - \boldsymbol{x}^{\boldsymbol{\alpha}''}| \\
&\leq \frac{1}{12\lambda^2} + \frac{|\boldsymbol{\alpha}'| - 1}{12\lambda^2} + \frac{|\boldsymbol{\alpha}''| - 1}{12\lambda^2} \\
&= \frac{|\boldsymbol{\alpha}| - 1}{12\lambda^2}.
\end{aligned}$$

$\square$

Next we prove Lemma 5.3 which can provide the coefficients of monomials in a Taylor polynomial at a point.

*Proof of Lemma 5.3.* The term $x^{\alpha}$ in the expansion of $\sum_{|\alpha| \leq \kappa} b_{\alpha}(x - a)^{\alpha}$ can only come from terms $(x - a)^{\nu}$ with $\nu \geq \alpha$ and $|\nu| \leq \kappa$. The coefficient of the term $x^{\alpha}$ in the expansion of $(x - a)^{\nu}$ is $\prod_{i=1}^{d} \binom{\nu_i}{\alpha_i} (-a_i)^{\nu_i - \alpha_i}$. By summing up coefficients from all terms $b_{\nu}(x - a)^{\nu}$ satisfying $\nu \geq \alpha$ and $|\nu| \leq \kappa$, we obtain that the coefficient of $x^{\alpha}$ is $\sum_{\substack{\nu \geq \alpha \\ |\nu| \leq \kappa}} b_{\nu} \prod_{i=1}^{d} \binom{\nu_i}{\alpha_i} (-a_i)^{\nu_i - \alpha_i}$. $\qquad\square$

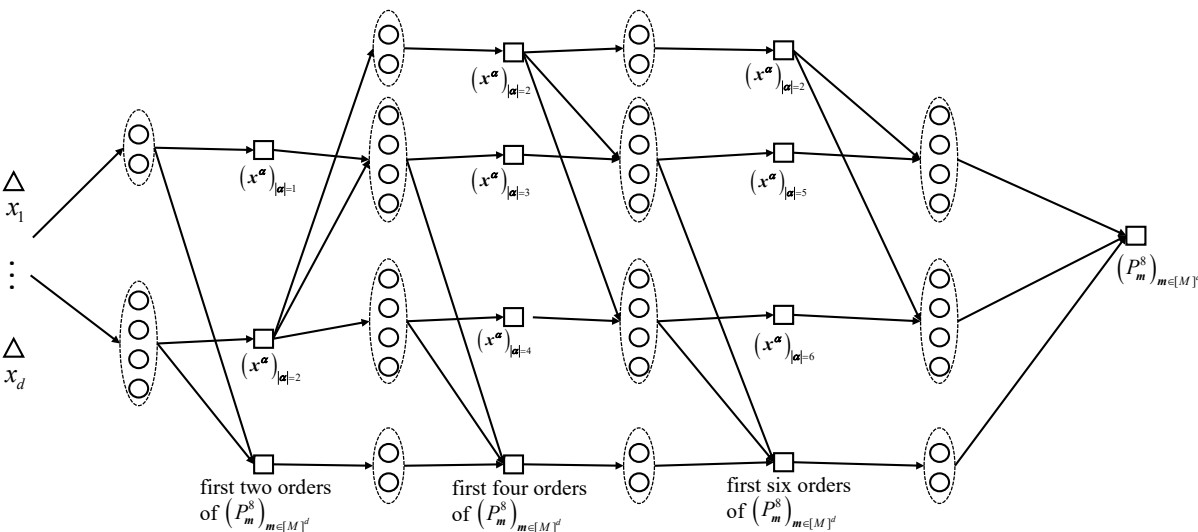

*Figure 3.* An example of constructive approximation for Taylor polynomials $(P^{\kappa}_{m})_{m \in [M]^d}$ by network $\mathcal{P}$ where $\kappa = 8$. "$\triangle$" stands for input neuron, "$\bigcirc$" stands for nonlinear neuron, and "$\square$" stands for linear neuron. The neurons are divided into several groups by the dashed ellipses. Except for the input neurons, the number of neuron marks does not represent the actual number of neurons. $(x^{\alpha})_{|\alpha|=n}$ refers to all monomials of order $n$ and $(P^{\kappa}_{m})_{m \in [M]^d}$ refers to Taylor polynomials of order $\kappa$ at all grid points.

**Lemma A.17** (Neural networks approximate $(P^{\kappa}_{m})_{m \in [M]^d}$)**.** *Let $\beta \in \mathbb{R}_{+}, \beta = \kappa + \gamma, \kappa \in \mathbb{N}, \gamma \in (0, 1]$, and $R \in \mathbb{R}$. For all $f \in \mathcal{C}^{\beta, R}([0, 1]^d), M \in \mathbb{N}_{+}$, and $\lambda \geq 2^{-\frac{1}{3}}$, letting $(P^{\kappa}_{m})_{m \in [M]^d}$ be $\kappa$th-order Taylor polynomials of $f$ at all grid points $\{m/M \mid m \in [M]^d\}$, there exists a low-rank Swish network $\mathcal{P} : [-1, 1]^d \to \mathbb{R}^{(M+1)^d}$ with depth*

$$\left\lceil \frac{\kappa}{2} \right\rceil, \tag{48}$$

*width of nonlinear layers*

$$2\binom{d+1}{d-1} + 4\binom{d+\kappa-2}{d-1} + 4\binom{d+\kappa-1}{d-1} + 2(M+1)^d, \tag{49}$$

*width of linear hidden layers*

$$\binom{d+1}{d-1} + \binom{d+\kappa-3}{d-1} + \binom{d+\kappa-2}{d-1} + (M+1)^d, \tag{50}$$

*upper bound of absolute values of parameters*

$$\max \left\{ 2\lambda^2 \max_{|\alpha| \leq \kappa} \left\{ \sum_{\substack{\nu \geq \alpha \\ |\nu| \leq \kappa}} \frac{R}{\nu!} \prod_{i=1}^{d} \binom{\nu_i}{\alpha_i} \right\}, 2\lambda^2 \right\}, \tag{51}$$

*and upper bound of number of nonzero parameters*

$$4\left\lceil\frac{\kappa}{2}\right\rceil\binom{d+1}{d-1} + 12\sum_{l=2}^{\kappa}\binom{d+l-1}{d-1} + (M+1)^d\left(2d + 4\left\lceil\frac{\kappa}{2}\right\rceil + 4\sum_{l=2}^{\kappa}\binom{d+l-1}{d-1}\right) \tag{52}$$

*such that*

$$\left\|\mathcal{P}(\boldsymbol{x}) - (P_{\boldsymbol{m}}^{\kappa}(\boldsymbol{x}))_{\boldsymbol{m}\in[M]^d}\right\|_{\infty} \leq \frac{1}{12\lambda^2}\sum_{2\leq|\boldsymbol{\alpha}|\leq\kappa}\left((|\boldsymbol{\alpha}|-1)\sum_{\substack{\boldsymbol{\nu}\geq\boldsymbol{\alpha}\\|\boldsymbol{\nu}|\leq\kappa}}\left(\frac{R}{\boldsymbol{\nu}!}\prod_{i=1}^{d}\binom{\nu_i}{\alpha_i}\right)\right) \tag{53}$$

*for all $\boldsymbol{x}\in[-1,1]^d$.*

*Proof of Lemma A.17.* Let $f\in\mathcal{C}^{\beta,R}([0,1]^d)$, $M\in\mathbb{N}_+$, and $\lambda\geq 2^{-\frac{1}{3}}$.

If $\beta\leq 1$, then $\kappa=0$. For any $\boldsymbol{m}\in[M]^d$, $P_{\boldsymbol{m}}^{\kappa}(x)=f(\boldsymbol{m}/M)$. Then we directly build a linear layer as the output layer without any connection to the input layer and with bias $(f(\boldsymbol{m}/M))_{\boldsymbol{m}\in[M]^d}$. We call the above network $\mathcal{P}$. Obviously, $\mathcal{P}(\boldsymbol{x})=(P_{\boldsymbol{m}}^{\kappa}(\boldsymbol{x}))_{\boldsymbol{m}\in[M]^d}$ for all $\boldsymbol{x}\in[-1,1]^d$ and $\mathcal{P}$ is of depth 0 (i.e. no hidden layer), number of nonzero parameters no more than $(M+1)^d$, and maximum absolute value of parameters no more than $R$.

If $\beta>1$, we construct a network layer by layer. In the following proof, we do not specifically mention the slight differences in the construction method when $\kappa$ is small or odd, but the way to handle these is quite naïve.

**Step 1: constructing the first nonlinear and linear layers**

By Lemma 5.6, for each input variable $x_i$, we construct two Swish neurons followed by one linear neuron to exactly preserve the value of $x_i$. We arrange neurons for preserving all $x_i$ in parallel and thus obtain a nonlinear layer of width $2d$ followed by a linear layer of width $d$. The total number of nonzero parameters of these two layers is $4d$ and the maximum absolute value of parameters of them is $1$.

Meanwhile we approximate each square term $x_i^2$ with two Swish neurons followed by one linear neuron by Lemma 5.4 and each cross term $x_i x_j$ with four Swish neurons followed by one linear neuron by Lemma 5.5. By arranging neurons for approximating all $x_i^2$ and $x_i x_j$, we obtain a nonlinear layer of width no more than $4\binom{d+1}{d-1}$ followed by a linear layer of width no more than $\binom{d+1}{d-1}$ because the number of all 2nd-order monomials is $\binom{d+1}{d-1}$. The total number of nonzero parameters of these two layers is no more than $12\binom{d+1}{d-1}$ and the maximum absolute value of parameters of them is $2\lambda^2$ since $\lambda\geq 2^{-\frac{1}{3}}$ implies $2\lambda^2\geq\frac{1}{\lambda}\geq\frac{1}{2\lambda}$. It is easy to know that given input $\boldsymbol{x}\in[-1,1]^d$ the outputs of the linear layer are equal to $\mathcal{M}_{\boldsymbol{\alpha}}(\boldsymbol{x})$ with $\boldsymbol{\alpha}=2$. So by Lemma A.16 the approximation error for 2nd-order monomials is bounded by $\frac{1}{12\lambda^2}$.

Next we construct a linear layer of width $(M+1)^d$ to approximate the first two orders of $(P_{\boldsymbol{m}}^{\kappa})_{\boldsymbol{m}\in[M]^d}$. For each $\boldsymbol{m}\in[M]^d$, the linear neuron, approximating the first two orders of $P_{\boldsymbol{m}}^{\kappa}$, links to two Swish neurons preserving $\boldsymbol{x}^{\boldsymbol{\alpha}}$ with weights $\sum_{\substack{\boldsymbol{\nu}\geq\boldsymbol{\alpha}\\|\boldsymbol{\nu}|\leq\kappa}}\frac{\partial^{\boldsymbol{\nu}}f(\boldsymbol{m}/M)}{\boldsymbol{\nu}!}\prod_{i=1}^{d}\binom{\nu_i}{\alpha_i}(-\frac{m_i}{M})^{\nu_i-\alpha_i}$ and $-\sum_{\substack{\boldsymbol{\nu}\geq\boldsymbol{\alpha}\\|\boldsymbol{\nu}|\leq\kappa}}\frac{\partial^{\boldsymbol{\nu}}f(\boldsymbol{m}/M)}{\boldsymbol{\nu}!}\prod_{i=1}^{d}\binom{\nu_i}{\alpha_i}(-\frac{m_i}{M})^{\nu_i-\alpha_i}$ for all $|\boldsymbol{\alpha}|=1$, two Swish neurons approximating the square term $\boldsymbol{x}^{\boldsymbol{\alpha}}$ with weights $2\lambda^2\sum_{\substack{\boldsymbol{\nu}\geq\boldsymbol{\alpha}\\|\boldsymbol{\nu}|\leq\kappa}}\frac{\partial^{\boldsymbol{\nu}}f(\boldsymbol{m}/M)}{\boldsymbol{\nu}!}\prod_{i=1}^{d}\binom{\nu_i}{\alpha_i}(-\frac{m_i}{M})^{\nu_i-\alpha_i}$ and $2\lambda^2\sum_{\substack{\boldsymbol{\nu}\geq\boldsymbol{\alpha}\\|\boldsymbol{\nu}|\leq\kappa}}\frac{\partial^{\boldsymbol{\nu}}f(\boldsymbol{m}/M)}{\boldsymbol{\nu}!}\prod_{i=1}^{d}\binom{\nu_i}{\alpha_i}(-\frac{m_i}{M})^{\nu_i-\alpha_i}$ for all $|\boldsymbol{\alpha}|=2$ satisfying $\boldsymbol{\alpha}=2\boldsymbol{\alpha}'$ where $|\boldsymbol{\alpha}'|=1$, and four Swish neurons approximating the cross term $\boldsymbol{x}^{\boldsymbol{\alpha}}$ with weights $2\lambda^2\sum_{\substack{\boldsymbol{\nu}\geq\boldsymbol{\alpha}\\|\boldsymbol{\nu}|\leq\kappa}}\frac{\partial^{\boldsymbol{\nu}}f(\boldsymbol{m}/M)}{\boldsymbol{\nu}!}\prod_{i=1}^{d}\binom{\nu_i}{\alpha_i}(-\frac{m_i}{M})^{\nu_i-\alpha_i}$, $2\lambda^2\sum_{\substack{\boldsymbol{\nu}\geq\boldsymbol{\alpha}\\|\boldsymbol{\nu}|\leq\kappa}}\frac{\partial^{\boldsymbol{\nu}}f(\boldsymbol{m}/M)}{\boldsymbol{\nu}!}\prod_{i=1}^{d}\binom{\nu_i}{\alpha_i}(-\frac{m_i}{M})^{\nu_i-\alpha_i}$, $-2\lambda^2\sum_{\substack{\boldsymbol{\nu}\geq\boldsymbol{\alpha}\\|\boldsymbol{\nu}|\leq\kappa}}\frac{\partial^{\boldsymbol{\nu}}f(\boldsymbol{m}/M)}{\boldsymbol{\nu}!}\prod_{i=1}^{d}\binom{\nu_i}{\alpha_i}(-\frac{m_i}{M})^{\nu_i-\alpha_i}$, and $-2\lambda^2\sum_{\substack{\boldsymbol{\nu}\geq\boldsymbol{\alpha}\\|\boldsymbol{\nu}|\leq\kappa}}\frac{\partial^{\boldsymbol{\nu}}f(\boldsymbol{m}/M)}{\boldsymbol{\nu}!}\prod_{i=1}^{d}\binom{\nu_i}{\alpha_i}(-\frac{m_i}{M})^{\nu_i-\alpha_i}$ for all $|\boldsymbol{\alpha}|=2$ satisfying $\boldsymbol{\alpha}=\boldsymbol{\alpha}'+\boldsymbol{\alpha}''$ where $|\boldsymbol{\alpha}'|=|\boldsymbol{\alpha}''|=1$ and $\boldsymbol{\alpha}'\neq\boldsymbol{\alpha}'$. In addition, to construct the 0th-order items, we add the bias term $\sum_{|\boldsymbol{\nu}|\leq\kappa}\frac{\partial^{\boldsymbol{\nu}}f(\boldsymbol{m}/M)}{\boldsymbol{\nu}!}\prod_{i=1}^{d}(-\frac{m_i}{M})^{\nu_i}$ for the linear neuron approximating the first two orders of $P_{\boldsymbol{m}}^{\kappa}$ for all $\boldsymbol{m}\in[M]^d$. Therefore the total number of nonzero parameters of this linear layer is no more than $(M+1)^d\left(2d+4\binom{d+1}{d-1}+1\right)$ and the absolute values of parameters are no more than $2\lambda^2\max_{|\boldsymbol{\alpha}|\leq 2}\sum_{\substack{\boldsymbol{\nu}\geq\boldsymbol{\alpha}\\|\boldsymbol{\nu}|\leq\kappa}}\frac{R}{\boldsymbol{\nu}!}\prod_{i=1}^{d}\binom{\nu_i}{\alpha_i}$. By Lemma A.16 and 5.3, we notice that our construction to the 0th- and 1st-order

terms is errorless, so the total approximation error to the first two orders of $(P^\kappa_{\boldsymbol{m}})_{\boldsymbol{m}\in[M]^d}$ only comes from the 2nd-order terms which is no more than $\frac{1}{12\lambda^2}\sum_{|\boldsymbol{\alpha}|=2}\sum_{\substack{\boldsymbol{\nu}\geq\boldsymbol{\alpha}\\|\boldsymbol{\nu}|\leq\kappa}}\frac{R}{\boldsymbol{\nu}!}\prod_{i=1}^d\binom{\nu_i}{\alpha_i}$.

Finally, we concatenate two nonlinear layers and three linear layers respectively, obtaining a nonlinear layer of width no more than $2d+4\binom{d+1}{d-1}$ followed by a linear layer of width no more than $d+\binom{d+1}{d-1}+(M+1)^d$. The total number of nonzero parameters of these two layers is no more than $4d+12\binom{d+1}{d-1}+(M+1)^d\left(2d+4\binom{d+1}{d-1}+1\right)$ and the absolute values of them are no more than $\max\left\{2\lambda^2,2\lambda^2\max_{|\boldsymbol{\alpha}|\leq2}\sum_{\substack{\boldsymbol{\nu}\geq\boldsymbol{\alpha}\\|\boldsymbol{\nu}|\leq\kappa}}\frac{R}{\boldsymbol{\nu}!}\prod_{i=1}^d\binom{\nu_i}{\alpha_i}\right\}$. The approximation error for the first two orders of $(P^\kappa_{\boldsymbol{m}})_{\boldsymbol{m}\in[M]^d}$ by the outputs of the last $(M+1)^d$ linear neurons is no more than $\frac{1}{12\lambda^2}\sum_{|\boldsymbol{\alpha}|=2}\sum_{\substack{\boldsymbol{\nu}\geq\boldsymbol{\alpha}\\|\boldsymbol{\nu}|\leq\kappa}}\frac{R}{\boldsymbol{\nu}!}\prod_{i=1}^d\binom{\nu_i}{\alpha_i}$.

**Step 2: constructing the second nonlinear and linear layers**

The method to construct the second nonlinear and linear layers is similar to the above.

We use a nonlinear layer of width $2\binom{d+1}{d-1}$ followed by a linear layer of width $\binom{d+1}{d-1}$ to preserve the approximate 2nd-order monomials constructed before. Meanwhile we multiply the 1st-order monomials with approximate 2nd-order monomials to construct approximate 3rd-order monomials using a nonlinear layer of width $4\binom{d+2}{d-1}$ followed by a linear layer of width $\binom{d+2}{d-1}$. In the meantime, we square approximate 2nd-order monomials and multiply different approximate 2nd-order monomials to approximate all 4th-order monomials using a nonlinear layer of width no more than $4\binom{d+3}{d-1}$ followed by a linear layer of width no more than $\binom{d+3}{d-1}$. Simultaneously we employ a nonlinear layer of width $2(M+1)^d$ followed by a linear layer of width $(M+1)^d$ to preserve the approximation of first two orders of $(P^\kappa_{\boldsymbol{m}})_{\boldsymbol{m}\in[M]^d}$, then connect the linear layer with the previous two nonlinear layers used to approximate 3rd- and 4th-order monomials to approximate the first four orders of $(P^\kappa_{\boldsymbol{m}})_{\boldsymbol{m}\in[M]^d}$.

Finally we concatenate four nonlinear layers and four linear layers respectively, obtaining a nonlinear layer of width no more than $2\binom{d+1}{d-1}+4\binom{d+2}{d-1}+4\binom{d+3}{d-1}+(M+1)^d$ followed by a linear layer of width no more than $\binom{d+1}{d-1}+\binom{d+2}{d-1}+\binom{d+3}{d-1}+(M+1)^d$. The total number of nonzero parameters of these two layers is no more than $4\binom{d+1}{d-1}+12\binom{d+2}{d-1}+12\binom{d+3}{d-1}+(M+1)^d\left(4+4\binom{d+2}{d-1}+4\binom{d+3}{d-1}\right)$ and the absolute values of them are no more than $\max\left\{2\lambda^2,2\lambda^2\max_{3\leq|\boldsymbol{\alpha}|\leq4}\sum_{\substack{\boldsymbol{\nu}\geq\boldsymbol{\alpha}\\|\boldsymbol{\nu}|\leq\kappa}}\frac{R}{\boldsymbol{\nu}!}\prod_{i=1}^d\binom{\nu_i}{\alpha_i}\right\}$. The approximation error for the first four orders of $(P^\kappa_{\boldsymbol{m}})_{\boldsymbol{m}\in[M]^d}$ by the outputs of the last $(M+1)^d$ linear neurons is no more than $\frac{1}{12\lambda^2}\sum_{2\leq|\boldsymbol{\alpha}|\leq4}(|\boldsymbol{\alpha}|-1)\sum_{\substack{\boldsymbol{\nu}\geq\boldsymbol{\alpha}\\|\boldsymbol{\nu}|\leq\kappa}}\frac{R}{\boldsymbol{\nu}!}\prod_{i=1}^d\binom{\nu_i}{\alpha_i}$.

**Step 3: constructing the $(l+1)$th nonlinear and linear layers by induction**

Let $l\in\mathbb{N}_+$ with $l\geq2$. Now suppose that we can directly obtain approximate 2rd-, $(2l-1)$th-, and $(2l)$th-order monomials and the first $2l$ orders of $(P^\kappa_{\boldsymbol{m}})_{\boldsymbol{m}\in[M]^d}$ from the last linear layer. The last nonlinear layer is of width no more than $2\binom{d+1}{d-1}+4\binom{d+2l-2}{d-1}+4\binom{d+2l-1}{d-1}+2(M+1)^d$ and the last linear layer is of width no more than $\binom{d+1}{d-1}+\binom{d+2l-2}{d-1}+\binom{d+2l-1}{d-1}+(M+1)^d$. The total number of nonzero parameters of them is no more than $4\binom{d+1}{d-1}+12\binom{d+2l-2}{d-1}+12\binom{d+2l-1}{d-1}+(M+1)^d\left(4+4\binom{d+2l-2}{d-1}+4\binom{d+2l-1}{d-1}\right)$ and the absolute values of parameters of them are no more than $\max\left\{2\lambda^2,2\lambda^2\max_{2l-1\leq|\boldsymbol{\alpha}|\leq2l}\sum_{\substack{\boldsymbol{\nu}\geq\boldsymbol{\alpha}\\|\boldsymbol{\nu}|\leq\kappa}}\frac{R}{\boldsymbol{\nu}!}\prod_{i=1}^d\binom{\nu_i}{\alpha_i}\right\}$. The approximation error for the first $2l$ orders of $(P^\kappa_{\boldsymbol{m}})_{\boldsymbol{m}\in[M]^d}$ by the outputs of the last $(M+1)^d$ linear neurons is no more than $\frac{1}{12\lambda^2}\sum_{2\leq|\boldsymbol{\alpha}|\leq2l}(|\boldsymbol{\alpha}|-1)\sum_{\substack{\boldsymbol{\nu}\geq\boldsymbol{\alpha}\\|\boldsymbol{\nu}|\leq\kappa}}\frac{R}{\boldsymbol{\nu}!}\prod_{i=1}^d\binom{\nu_i}{\alpha_i}$.

Then, following the above way, we use a nonlinear layer of width $2\binom{d+1}{d-1}$ followed by a linear layer of width $\binom{d+1}{d-1}$ to preserve the approximate 2nd-order monomials. Meanwhile we multiply approximate 2nd-order monomials with approximate $(2l-1)$th-order monomials to construct approximate $(2l+1)$th-order monomials using a nonlinear layer of width $4\binom{d+2l}{d-1}$ followed by a linear layer of width $\binom{d+2l}{d-1}$ and multiply approximate 2nd-order monomials with approximate $(2l)$th-order monomials to construct approximate $(2l+2)$th-order monomials using a nonlinear layer of width $4\binom{d+2l+1}{d-1}$ followed by a linear layer of width $\binom{d+2l+1}{d-1}$. Simultaneously we employ a nonlinear layer of width $2(M+1)^d$ followed by a linear layer of width $(M+1)^d$ to preserve the approximation of first $(2l)$ orders of $(P^\kappa_{\boldsymbol{m}})_{\boldsymbol{m}\in[M]^d}$, then connect the linear

layer with the previous two nonlinear layers used to approximate $(2l+1)$th- and $(2l+2)$th-order monomials to approximate the first $2l+2$ orders of $(P_{\boldsymbol{m}}^{\kappa})_{\boldsymbol{m}\in[M]^d}$.

Finally we concatenate these four nonlinear layers and four linear layers in parallel respectively, obtaining a nonlinear layer of width no more than $2\binom{d+1}{d-1} + 4\binom{d+2l}{d-1} + 4\binom{d+2l+1}{d-1} + (M+1)^d$ followed by a linear layer of width no more than $\binom{d+1}{d-1} + \binom{d+2l}{d-1} + \binom{d+2l+1}{d-1} + (M+1)^d$. The total number of nonzero parameters of these two layers is no more than $4\binom{d+1}{d-1} + 12\binom{d+2l}{d-1} + 12\binom{d+2l+1}{d-1} + (M+1)^d\left(4 + 4\binom{d+2l}{d-1} + 4\binom{d+2l+1}{d-1}\right)$ and the absolute values of them are no more than $\max\left\{2\lambda^2, 2\lambda^2 \max_{2l+1\leq|\boldsymbol{\alpha}|\leq2l+2} \sum_{\substack{\boldsymbol{\nu}\geq\boldsymbol{\alpha}\\|\boldsymbol{\nu}|\leq\kappa}} \frac{R}{\boldsymbol{\nu}!}\prod_{i=1}^d\binom{\nu_i}{\alpha_i}\right\}$. The approximation error for the first $2l+2$ orders of $(P_{\boldsymbol{m}}^{\kappa})_{\boldsymbol{m}\in[M]^d}$ by the outputs of the last $(M+1)^d$ linear neurons is no more than $\frac{1}{12\lambda^2}\sum_{2\leq|\boldsymbol{\alpha}|\leq2l+2}(|\boldsymbol{\alpha}| - 1)\sum_{\substack{\boldsymbol{\nu}\geq\boldsymbol{\alpha}\\|\boldsymbol{\nu}|\leq\kappa}} \frac{R}{\boldsymbol{\nu}!}\prod_{i=1}^d\binom{\nu_i}{\alpha_i}$.

Through the above process, we finally construct a network of depth $\lceil\frac{\kappa}{2}\rceil$, called $\mathcal{P}$, to approximate Taylor polynomials at all grid points, $(P_{\boldsymbol{m}}^{\kappa})_{\boldsymbol{m}\in[M]^d}$, with error no more than

$$\frac{1}{12\lambda^2}\sum_{2\leq|\boldsymbol{\alpha}|\leq\kappa}(|\boldsymbol{\alpha}| - 1)\sum_{\substack{\boldsymbol{\nu}\geq\boldsymbol{\alpha}\\|\boldsymbol{\nu}|\leq\kappa}} \frac{R}{\boldsymbol{\nu}!}\prod_{i=1}^d\binom{\nu_i}{\alpha_i}.$$

Considering that there is no need to construct 2nd-, $(2\lceil\frac{\kappa}{2}\rceil - 1)$th-, and $(2\lceil\frac{\kappa}{2}\rceil)$th-order monomials at the last linear layer, the maximum width of nonlinear layers is no more than $2\binom{d+1}{d-1} + 4\binom{d+\kappa-2}{d-1} + 4\binom{d+\kappa-1}{d-1} + 2(M+1)^d$, the maximum width of linear layers no more than $\binom{d+1}{d-1} + \binom{d+\kappa-3}{d-1} + \binom{d+\kappa-2}{d-1} + (M+1)^d$, the absolute values of parameters are no more than $\max\left\{2\lambda^2 \max_{|\boldsymbol{\alpha}|\leq\kappa}\left\{\sum_{\substack{\boldsymbol{\nu}\geq\boldsymbol{\alpha}\\|\boldsymbol{\nu}|\leq\kappa}} \frac{R}{\boldsymbol{\nu}!}\prod_{i=1}^d\binom{\nu_i}{\alpha_i}\right\}, 2\lambda^2\right\}$, and the number of nonzero parameters is no more than $4d + 4\left(\lceil\frac{\kappa}{2}\rceil - 2\right)\binom{d+1}{d-1} + 12\sum_{l=2}^{\kappa}\binom{d+l-1}{d-1} + (M+1)^d\left(1 + 2d + 4\left(\lceil\frac{\kappa}{2}\rceil - 1\right) + 4\sum_{l=2}^{\kappa}\binom{d+l-1}{d-1}\right)$. $\qquad\square$

## A.3. Approximating $(\phi_{\boldsymbol{m}}^{\tau})_{\boldsymbol{m}\in[M]^d}$ by a Low-Rank Swish Network $\mathcal{G}$

We first define a series of functions $prod_r$ where $r \in \mathbb{N}_+$ by stacking $mult$, then analyze the error of approximating $(x_1,\ldots,x_r)^{\top} \mapsto \prod_{i=1}^r x_i$ using $prod_r$. When constructing $(\phi_{\boldsymbol{m}}^{\tau})_{\boldsymbol{m}\in[M]^d}$ in the proof of Lemma A.22, we can see that $prod_r$ is implemented by hidden neurons of network $\mathcal{G}$ for $r \in [d]_+$. For all $r \in \mathbb{N}_+$, the function $prod_r : \mathbb{R}^r \to \mathbb{R}$ is defined by

1. when $r = 1$, $prod_1(x) := x$;

2. when $r \geq 2$, $prod_r(x_1,\ldots,x_r) := mult(prod_{2^q}(x_1,\ldots,x_{2^q}), prod_{r-2^q}(x_{2^q+1},\ldots,x_r))$ where $q \in \mathbb{N}$ and $2^q < r \leq 2^{q+1}$.

**Lemma A.18** (Boundedness of $prod_r$). *Let $r \in \mathbb{N}_+$. For all $\boldsymbol{x} \in \mathbb{R}^r$,*

$$|prod_r(\boldsymbol{x})| \leq \prod_{i=1}^r |x_i|. \tag{54}$$

*Proof of Lemma A.18.* We prove this lemma by induction. When $r = 1$, for all $x \in \mathbb{R}$,

$$|prod_1(x)| = |x|.$$

Assume that $\forall \boldsymbol{x} \in \mathbb{R}^s, |prod_s(\boldsymbol{x})| \leq \prod_{i=1}^s |x_i|$ holds for all $s \leq r$. Then, letting $q \in \mathbb{N}$ satisfying $2^q < r+1 \leq 2^{q+1}$, for

all $\boldsymbol{x} \in \mathbb{R}^{r+1}$,

$$
\begin{aligned}
|prod_{r+1}(\boldsymbol{x})| &= |mult(prod_{2^q}(x_1, \ldots, x_{2^q}), prod_{r-2^q}(x_{2^q+1}, \ldots, x_{r+1}))| \\
&\leq |prod_{2^q}(x_1, \ldots, x_{2^q}) \cdot prod_{r+1-2^q}(x_{2^q+1}, \ldots, x_{r+1})| \quad \text{(by Lemma A.14)} \\
&\leq \prod_{i=1}^{2^q} |x_i| \cdot \prod_{i=2^q+1}^{r+1} |x_i| \quad \text{(by induction)} \\
&= \prod_{i=1}^{r+1} |x_i|.
\end{aligned}
$$

$\square$

**Lemma A.19.** *Let $r \in \mathbb{N}_+$ and $\lambda \in \mathbb{R}_+$. For all $\boldsymbol{x} \in [-1, 1]^r$,*

$$
\left| prod_r(\boldsymbol{x}) - \prod_{i=1}^{r} x_i \right| \leq \frac{r-1}{12\lambda^2}. \tag{55}
$$

*Proof of Lemma A.19.* We prove it by induction. When $r = 1$, for all $x \in [-1, 1]$,

$$
|prod_1(x) - x| = 0. \tag{56}
$$

Assume that $\forall \boldsymbol{x} \in [-1, 1]^s$, $|prod_r(\boldsymbol{x}) - \prod_{i=1}^{s} x_i| \leq \frac{s-1}{12\lambda^2}$ holds for all $s \leq r$. Then, letting $q \in \mathbb{N}$ satisfying $2^q < r+1 \leq 2^{q+1}$, for all $\boldsymbol{x} \in \mathbb{R}^{r+1}$, by Lemma 5.5 and A.18,

$$
\left| prod_{r+1}(\boldsymbol{x}) - \prod_{i=1}^{r+1} x_i \right|
$$

$$
\leq |mult(prod_{2^q}(x_1, \ldots, x_{2^q}), prod_{r+1-2^q}(x_{2^q+1}, \ldots, x_{r+1})) - prod_{2^q}(x_1, \ldots, x_{2^q}) \cdot prod_{r+1-2^q}(x_{2^q+1}, \ldots, x_{r+1})| +
$$

$$
\left| prod_{2^q}(x_1, \ldots, x_{2^q}) \cdot prod_{r+1-2^q}(x_{2^q+1}, \ldots, x_{r+1}) - \prod_{i=1}^{r+1} x_i \right|
$$

$$
\leq \frac{1}{12\lambda^2} + \left| prod_{2^q}(x_1, \ldots, x_{2^q}) - \prod_{i=1}^{2^q} x_i \right| + \left| prod_{r+1-2^q}(x_1, \ldots, x_{2^q}) - \prod_{i=2^q+1}^{r} x_i \right|
$$

$$
\leq \frac{1}{12\lambda^2} + \frac{2^q - 1}{12\lambda^2} + \frac{r+1-2^q-1}{12\lambda^2} \quad \text{(by induction)}
$$

$$
\leq \frac{r-1}{12\lambda^2}.
$$

$\square$

Next we introduce several lemmas which are helpful to explain how to approximate $(\phi_{\boldsymbol{m}}^{\tau})_{\boldsymbol{m} \in [M]^d}$ using a network.

**Lemma A.20.** *For all $n \in \mathbb{N}_+$,*

$$
2^n \geq 2n. \tag{57}
$$

*Proof of .* We prove it by induction. When $n = 1$, $2^1 = 2 \cdot 1$. $\forall n \in \mathbb{N}_+$, if $2^n \geq 2n$, then

$$
2^{n+1} = 2 \cdot 2^n \geq 2 \cdot 2n = 2(n + n) \geq 2(n + 1).
$$

$\square$

**Lemma A.21.** *For all $d, M \in \mathbb{N}_+$ and $q \in \mathbb{N}$, if $d \geq 2^q$, then*

$$
(M + 1)^d \geq (M + 1)^{2^q} \left\lceil \frac{d}{2^q} \right\rceil. \tag{58}
$$

*Proof of Lemma A.21.* There exists $k \in \mathbb{N}_+$ and $p \in \mathbb{N}$ with $0 \le p < 2^q$ such that $d = k2^q + p$. If $p = 0$, then by Lemma A.20,

$$\left\lceil \frac{d}{2^q} \right\rceil = k \le 2^{k-1} \le (M+1)^{2^q(k-1)} \quad \Rightarrow \quad (M+1)^{2^q} \left\lceil \frac{d}{2^q} \right\rceil \le (M+1)^{k2^q} = (M+1)^d.$$

If $p > 0$, then by Lemma A.20,

$$\left\lceil \frac{d}{2^q} \right\rceil = k + 1 \le 2k \le 2 \cdot 2^{k-1} \le 2(M+1)^{2^q(k-1)} \le (M+1)^{2^q(k-1)+p}$$

$$\Rightarrow \quad (M+1)^{2^q} \left\lceil \frac{d}{2^q} \right\rceil \le (M+1)^{k2^q+p} = (M+1)^d.$$

$\square$

**Lemma A.22** (Neural networks approximates $(\phi_{\boldsymbol{m}}^\tau)_{\boldsymbol{m} \in [M]^d}$)**.** *For all $M \in \mathbb{N}_+, \lambda \ge 2^{-\frac{2}{3}}$, and $\tau \ge 1$, there exists a low-rank Swish network $\mathcal{G} : [-1,1]^d \to \mathbb{R}^{(M+1)^d}$ with depth*

$$\lceil \log_2 d \rceil + 1, \tag{59}$$

*width of nonlinear layers*

$$4(M+1)^d, \tag{60}$$

*width of linear layers*

$$(M+1)^d, \tag{61}$$

*upper bound of absolute values of parameters*

$$\max\{(3M+2)\tau, \; 2\lambda^2\}, \tag{62}$$

*and upper bound of number of nonzero parameters*

$$(12 \lceil \log_2 d \rceil + 8)(M+1)^d, \tag{63}$$

*such that*

$$\|\mathcal{G}(\boldsymbol{x}) - (\phi_{\boldsymbol{m}}^\tau(\boldsymbol{x}))_{\boldsymbol{m} \in [M]^d}\|_\infty \le \frac{d-1}{12\lambda^2} \tag{64}$$

*for all $\boldsymbol{x} \in [-1,1]^d$.*

*Proof of Lemma A.22.* For any $i \in \{1, 2, \ldots, d\}$, we list $\psi^\tau \left(3M \left(x_i - \frac{m_i}{M}\right)\right)$ for all $m_i \in \{0, 1, \ldots, M\}$:

$$\psi^\tau \left(3M \left(x_i - \frac{0}{M}\right)\right) = \frac{1}{\tau}(\rho(3M\tau x_i + 2\tau) - \rho(3M\tau x_i + \tau) - \rho(3M\tau x_i - \tau) + \rho(3M\tau x_i - 2\tau)),$$

$$\psi^\tau \left(3M \left(x_i - \frac{1}{M}\right)\right) = \frac{1}{\tau}(\rho(3M\tau x_i - \tau) - \rho(3M\tau x_i - 2\tau) - \rho(3M\tau x_i - 4\tau) + \rho(3M\tau x_i - 5\tau)),$$

$$\psi^\tau \left(3M \left(x_i - \frac{2}{M}\right)\right) = \frac{1}{\tau}(\rho(3M\tau x_i - 4\tau) - \rho(3M\tau x_i - 5\tau) - \rho(3M\tau x_i - 7\tau) + \rho(3M\tau x_i - 8\tau)),$$

$$\cdots \cdots$$

$$\psi^\tau \left(3M \left(x_i - \frac{M-1}{M}\right)\right) = \frac{1}{\tau}(\rho(3M\tau x_i + (-3M+5)\tau) - \rho(3M\tau x_i + (-3M+4)\tau)$$
$$- \rho(3M\tau x_i + (-3M+2)\tau) + \rho(3M\tau x_i + (-3M+1)\tau)),$$

$$\psi^\tau \left(3M \left(x_i - \frac{M}{M}\right)\right) = \frac{1}{\tau}(\rho(3M\tau x_i + (-3M+2)\tau) - \rho(3M\tau x_i + (-3M+1)\tau)$$
$$- \rho(3M\tau x_i + (-3M-1)\tau) + \rho(3M\tau x_i + (-3M-2)\tau)).$$

As we can see, to exactly construct $\psi^\tau(3M(x_i - \frac{m_i}{M}))$ for all $i \in [d]_+$ and $m_i \in [M]$, we only need a nonlinear layer of width $(2M+4)d$ followed by a linear layer of width $(M+1)d$, where $(2M+4)d \le 4(M+1)d \le 4(M+1)^d$ and $(M+1)d \le (M+1)^d$ by Lemma A.21 with $q = 0$. The number of nonzero parameters of these two layers is no more than $2(2M+4)d + 4(M+1)d \le 8(M+1)d \le 8(M+1)^d$ (by Lemma A.21 with $q = 0$) and the absolute values of parameters of them are no more than $(3M+2)\tau$ since $\tau \ge 1$ implies $(3M+2)\tau \ge 1/\tau$.

Let $q \in \mathbb{N}$. There exist $k \in \mathbb{N}$ and $p \in \mathbb{N}$ with $0 \le p < 2^q$ such that $d = k2^q + p$. Assume that there is a network of depth $q+1$, width of nonlinear layers $4(M+1)^d$, width of linear layers $(M+1)^d$, upper bound of number of nonzero parameters $(12q+8)(M+1)^d$, and upper bound of absolute values of parameters $\max\{(3M+2)\tau, 2\lambda^2\}$ outputting

$$prod_{2^q}\left(\psi^\tau\left(3M\left(x_j - \frac{m_j}{M}\right)\right), \ldots, \psi^\tau\left(3M\left(x_{j+2^q-1} - \frac{m_{j+2^q-1}}{M}\right)\right)\right)$$

to approximate

$$\prod_{\iota=0}^{2^q-1} \psi^\tau\left(3M\left(x_{j+\iota} - \frac{m_{j+\iota}}{M}\right)\right)$$

for all $j \in \{1, 2^q+1, 2\cdot 2^q+1, \ldots, (k-1)\cdot 2^q+1\}$ and $\boldsymbol{m} \in [M]^d$ if $k > 0$ (i.e. $d \ge 2^q$) and

$$prod_p\left(\psi^\tau\left(3M\left(x_{d-p+1} - \frac{m_{d-p+1}}{M}\right)\right), \ldots, \psi^\tau\left(3M\left(x_d - \frac{m_d}{M}\right)\right)\right)$$

to approximate

$$\prod_{\iota=-p+1}^{0} \psi^\tau\left(3M\left(x_{d+\iota} - \frac{m_{d+\iota}}{M}\right)\right)$$

for all $\boldsymbol{m} \in [M]^d$ if $p > 0$.

Then when $q' = q + 1$, there exist $k' \in \mathbb{N}$ and $p' \in \mathbb{N}$ with $0 \le p' < 2^{q'}$ such that $d = k'2^{q'} + p'$. We build a nonlinear layer of width $4(M+1)^{2^{q'}}\lceil\frac{d}{2^{q'}}\rceil \le 4(M+1)^d$ and a subsequent linear layer of width $(M+1)^{2^{q'}}\lceil\frac{d}{2^{q'}}\rceil \le (M+1)^d$ upon the network to approximately multiply the outputs of it using $mult$. Thus the new network outputs

$$prod_{2^{q'}}\left(\psi^\tau\left(3M\left(x_j - \frac{m_j}{M}\right)\right), \ldots, \psi^\tau\left(3M\left(x_{j+2^{q'}-1} - \frac{m_{j+2^{q'}-1}}{M}\right)\right)\right)$$

to approximate

$$\prod_{\iota=0}^{2^{q'}-1} \psi^\tau\left(3M\left(x_{j+\iota} - \frac{m_{j+\iota}}{M}\right)\right)$$

within $\frac{2^{q'}-1}{12\lambda^2}$ (by Lemma A.19) for all $j \in \{1, 2^{q'}+1, 2\cdot 2^{q'}+1, \ldots, (k'-1)\cdot 2^{q'}+1\}$ and $\boldsymbol{m} \in [M]^d$ if $k' > 0$ (i.e. $d \ge 2^{q'}$) and

$$prod_{p'}\left(\psi^\tau\left(3M\left(x_{d-p'+1} - \frac{m_{d-p'+1}}{M}\right)\right), \ldots, \psi^\tau\left(3M\left(x_d - \frac{m_d}{M}\right)\right)\right)$$

to approximate

$$\prod_{\iota=-p'+1}^{0} \psi^\tau\left(3M\left(x_{d+\iota} - \frac{m_{d+\iota}}{M}\right)\right)$$

within $\frac{p'-1}{12\lambda^2}$ (by Lemma A.19) for all $\boldsymbol{m} \in [M]^d$ if $p' > 0$. Conbining the newly built two layers, the new network is of depth $q'+1$, width of nonlinear layer $4(M+1)^d$, width of linear layer $(M+1)^d$, upper bound of number of nonzero parameters $(12q'+8)(M+1)^d$, and upper bound of absolute values of parameters $\max\{(3M+2)\tau, 2\lambda^2\}$. And the approximation error is within $\frac{d-1}{12\lambda^2}$.

We stop the above construction process after the iteration with $q = \lceil\log_2 d\rceil$ because at this point, $(\phi_{\boldsymbol{m}}^\tau)_{\boldsymbol{m}\in[M]^d}$ is just approximately constructed.

$\square$

## A.4. Approximating $\sum_{\boldsymbol{m}\in[M]^d} P_{\boldsymbol{m}}^\kappa \phi_{\boldsymbol{m}}^\tau$ by the Inner Product of $\mathcal{P}$ and $\mathcal{G}$

We denote the output dimension of $\mathcal{P}$ to approximate $P_{\boldsymbol{m}}^\kappa$ by $\mathcal{P}_{\boldsymbol{m}}$ and the output dimension of $\mathcal{G}$ to approximate $\phi_{\boldsymbol{m}}^\tau$ by $\mathcal{G}_{\boldsymbol{m}}$. Here we introduce some inequalities about $\mathcal{P}_{\boldsymbol{m}}$ and $\mathcal{G}_{\boldsymbol{m}}$.

**Lemma A.23** (Boundedness of $\mathcal{P}_{\boldsymbol{m}}$ and $\mathcal{G}_{\boldsymbol{m}}$). *Let $M \in \mathbb{N}_+$, $\tau \geq 1$, and $\lambda \geq 2^{-\frac{1}{3}}$. For all $\boldsymbol{x} \in [0,1]^d$ and $\boldsymbol{m} \in [M]^d$,*

$$|\mathcal{P}_{\boldsymbol{m}}(\boldsymbol{x})| \leq \frac{1}{6\sqrt[3]{2}} \sum_{2 \leq |\boldsymbol{\alpha}| \leq \kappa} \left( (|\boldsymbol{\alpha}| - 1) \sum_{\substack{\boldsymbol{\nu} \geq \boldsymbol{\alpha} \\ |\boldsymbol{\nu}| \leq \kappa}} \left( \frac{R}{\boldsymbol{v}!} \prod_{i=1}^d \binom{\nu_i}{\alpha_i} \right) \right) + \binom{\kappa + d - 1}{d - 1} R + R \tag{65}$$

*and*

$$|\mathcal{G}_{\boldsymbol{m}}(\boldsymbol{x})| \leq \frac{d - 1}{6\sqrt[3]{2}} + (2\|\rho'\|_\infty)^d. \tag{66}$$

*Proof of A.23.* To show the first inequality, by Lemma A.17 and A.1,

$$|\mathcal{P}_{\boldsymbol{m}}(\boldsymbol{x})| \leq |\mathcal{P}_{\boldsymbol{m}}(\boldsymbol{x}) - P_{\boldsymbol{m}}^\kappa(\boldsymbol{x})| + |P_{\boldsymbol{m}}^\kappa(\boldsymbol{x})|$$

$$\leq \frac{1}{12\lambda^2} \sum_{2 \leq |\boldsymbol{\alpha}| \leq \kappa} \left( (|\boldsymbol{\alpha}| - 1) \sum_{\substack{\boldsymbol{\nu} \geq \boldsymbol{\alpha} \\ |\boldsymbol{\nu}| \leq \kappa}} \left( \frac{R}{\boldsymbol{v}!} \prod_{i=1}^d \binom{\nu_i}{\alpha_i} \right) \right) + \binom{\kappa + d - 1}{d - 1} R + R$$

$$\leq \frac{1}{6\sqrt[3]{2}} \sum_{2 \leq |\boldsymbol{\alpha}| \leq \kappa} \left( (|\boldsymbol{\alpha}| - 1) \sum_{\substack{\boldsymbol{\nu} \geq \boldsymbol{\alpha} \\ |\boldsymbol{\nu}| \leq \kappa}} \left( \frac{R}{\boldsymbol{v}!} \prod_{i=1}^d \binom{\nu_i}{\alpha_i} \right) \right) + \binom{\kappa + d - 1}{d - 1} R + R.$$

To show the second inequality, by Lemma A.22 and A.4,

$$|\mathcal{G}_{\boldsymbol{m}}(\boldsymbol{x})| \leq |\mathcal{G}_{\boldsymbol{m}}(\boldsymbol{x}) - \phi_{\boldsymbol{m}}^\tau(\boldsymbol{x})| + |\phi_{\boldsymbol{m}}^\tau(\boldsymbol{x})|$$

$$\leq \frac{d - 1}{12\lambda^2} + (2\|\rho'\|_\infty)^d$$

$$\leq \frac{d - 1}{6\sqrt[3]{2}} + (2\|\rho'\|_\infty)^d.$$

$\square$

**Theorem A.24** (Neural Networks Approximates $f \in \mathcal{C}^{\beta, R}([0, 1]^d)$). *Let $\beta \in \mathbb{R}_+, \beta = \kappa + \gamma, \kappa \in \mathbb{N}, \gamma \in (0, 1]$, and $R \in \mathbb{R}_+$. For all $f \in \mathcal{C}^{\beta, R}([0, 1]^d)$, $M \in \mathbb{N}_+$, $\lambda \geq 2^{-\frac{1}{3}}$, and $\tau \geq 1$, there exists a low-rank Swish network $nn : [0, 1]^d \to \mathbb{R}$ with depth*

$$\max\left\{ \left\lceil \frac{\kappa}{2} \right\rceil, \lceil \log_2 d \rceil + 1 \right\} + 1, \tag{67}$$

*width of nonlinear layers*

$$2\binom{d + 1}{d - 1} + 4\binom{d + \kappa - 2}{d - 1} + 4\binom{d + \kappa - 1}{d - 1} + 6(M + 1)^d, \tag{68}$$

*width of linear hidden layers*

$$\binom{d + 1}{d - 1} + \binom{d + \kappa - 3}{d - 1} + \binom{d + \kappa - 2}{d - 1} + 2(M + 1)^d, \tag{69}$$

*upper bound of number of nonzero parameters*

$$4\left\lceil \frac{\kappa}{2} \right\rceil \binom{d + 1}{d - 1} + 12 \sum_{l=2}^\kappa \binom{d + l - 1}{d - 1} + $$
$$(M + 1)^d \left( 24 + 16\lceil \log_2 d \rceil + 2d + 8\left\lceil \frac{\kappa}{2} \right\rceil + 4 \sum_{l=2}^\kappa \binom{d + l - 1}{d - 1} \right) \tag{70}$$

*and upper bound of absolute values of parameters*

$$\max\left\{(3M+2)\tau,\ 2\lambda^2\max_{|\boldsymbol{\alpha}|\leq\kappa}\left\{\sum_{\substack{\boldsymbol{\nu}\geq\boldsymbol{\alpha}\\|\boldsymbol{\nu}|\leq\kappa}}\frac{R}{\boldsymbol{v}!}\prod_{i=1}^{d}\binom{\nu_i}{\alpha_i}\right\},\ 2\lambda^2\right\},\tag{71}$$

*such that*

$$|nn(\boldsymbol{x})-f(\boldsymbol{x})|$$
$$\leq\frac{(M+1)^d}{12\lambda^2}\left(\frac{C_1^4+6C_1^2C_2^2+C_2^4}{8}+C_2C_3+C_4(d-1)\right)+\tau e^{-\tau}C_5+M^{-\beta}C_6+(M+1)^d\tau e^{-\tau}C_7$$
$$\leq\frac{(M+1)^d}{12\lambda^2}\left(\frac{C_1^4+6C_1^2C_2^2+C_2^4}{8}+C_2C_3+C_4(d-1)\right)+M^{-\beta}C_6+(M+1)^d\tau e^{-\tau}\left(\frac{C_5}{2^d}+C_7\right)$$

*for all* $\boldsymbol{x}\in[0,1]^d$, *where*

$$C_1:=\frac{1}{6\sqrt[3]{2}}\sum_{2\leq|\boldsymbol{\alpha}|\leq\kappa}\left((|\boldsymbol{\alpha}|-1)\sum_{\substack{\boldsymbol{\nu}\geq\boldsymbol{\alpha}\\|\boldsymbol{\nu}|\leq\kappa}}\left(\frac{R}{\boldsymbol{v}!}\prod_{i=1}^{d}\binom{\nu_i}{\alpha_i}\right)\right)+\binom{\kappa+d-1}{d-1}R+R\,,$$

$$C_2:=\frac{d-1}{6\sqrt[3]{2}}+(2\|\rho'\|_\infty)^d\,,$$

$$C_3:=\sum_{2\leq|\boldsymbol{\alpha}|\leq\kappa}\left((|\boldsymbol{\alpha}|-1)\sum_{\substack{\boldsymbol{\nu}\geq\boldsymbol{\alpha}\\|\boldsymbol{\nu}|\leq\kappa}}\left(\frac{R}{\boldsymbol{v}!}\prod_{i=1}^{d}\binom{\nu_i}{\alpha_i}\right)\right)\,,$$

$$C_4:=\binom{\kappa+d-1}{d-1}R\left\|\boldsymbol{x}-\frac{\boldsymbol{m}}{M}\right\|_\infty^\beta+R\,,$$

$$C_5:=6R\frac{(2\|\rho'\|_\infty)^d-1}{2\|\rho'\|_\infty-1}\,,$$

$$C_6:=3^d\binom{\kappa+d-1}{d-1}R(2\|\rho'\|_\infty)^d\,,$$

$$C_7=6\binom{\kappa+d-1}{d-1}R(2\|\rho'\|_\infty)^{d-1}\,.$$

*Proof of Theorem A.24.* For all $f\in\mathcal{C}^{\beta,R}([0,1]^d)$, $M\in\mathbb{N}_+$, $\lambda\geq 2^{-\frac{1}{3}}$, and $\tau\geq 1$, by Lemma A.17 and A.22, there exist network $\mathcal{P}$ and $\mathcal{G}$ approximating $(P_{\boldsymbol{m}}^\kappa)_{\boldsymbol{m}\in[M]^d}$ and $(\phi_{\boldsymbol{m}}^\tau)_{\boldsymbol{m}\in[M]^d}$ respectively. Considering that the depths of $\mathcal{P}$ and $\mathcal{G}$ may be not identical, we construct several nonlinear and linear layers of width $2(M+1)^d$ and $(M+1)^d$, which mimic the identity function by Lemma 5.6, upon the shallow one to align their depths. Note that adding these layers does not change the output, width, and upper bound of absolute values of parameters of the shallow network, but its number of nonzero parameters increases

$$4\left(\max\left\{\left\lceil\frac{\kappa}{2}\right\rceil,\lceil\log_2 d\rceil+1\right\}-\min\left\{\left\lceil\frac{\kappa}{2}\right\rceil,\lceil\log_2 d\rceil+1\right\}\right)(M+1)^d\leq 4\left(\left\lceil\frac{\kappa}{2}\right\rceil+\lceil\log_2 d\rceil+1\right)(M+1)^d.$$

Define a function $nn:[0,1]^d\to\mathbb{R}$ as the inner product of $\mathcal{P}$ and $\mathcal{G}$, i.e.

$$nn(\boldsymbol{x}):=\sum_{\boldsymbol{m}\in[M]^d}mult\left(\mathcal{P}_{\boldsymbol{m}}(\boldsymbol{x}),\mathcal{G}_{\boldsymbol{m}}(\boldsymbol{x})\right).\tag{72}$$

The function $nn$ can be implemented by adding a nonlinear layer of width $4(M+1)^d$ and a subsequent linear layer of width 1 upon depth-aligned $\mathcal{P}$ and $\mathcal{G}$ to approximately multiply $\mathcal{P}_{\boldsymbol{m}}$ and $\mathcal{P}_{\boldsymbol{m}}$ for all $\boldsymbol{m}\in[M]^d$ and sup them up. The number of additional nonzero parameters is $12(M+1)^d$.

Next we analyze the approximation error: for all $\boldsymbol{x} \in [0,1]^d$,

$$|nn(\boldsymbol{x}) - f(\boldsymbol{x})|$$

$$\leq \left| \sum_{\boldsymbol{m} \in [M]^d} mult\left(\mathcal{P}_{\boldsymbol{m}}(\boldsymbol{x}), \mathcal{G}_{\boldsymbol{m}}(\boldsymbol{x})\right) - \sum_{\boldsymbol{m} \in [M]^d} \mathcal{P}_{\boldsymbol{m}}(\boldsymbol{x})\mathcal{G}_{\boldsymbol{m}}(\boldsymbol{x}) \right| + \left| \sum_{\boldsymbol{m} \in [M]^d} \mathcal{P}_{\boldsymbol{m}}(\boldsymbol{x})\mathcal{G}_{\boldsymbol{m}}(\boldsymbol{x}) - \sum_{\boldsymbol{m} \in [M]^d} P_{\boldsymbol{m}}^\kappa(\boldsymbol{x})\phi_{\boldsymbol{m}}^\tau(\boldsymbol{x}) \right| +$$

$$\left| \sum_{\boldsymbol{m} \in [M]^d} P_{\boldsymbol{m}}^\kappa(\boldsymbol{x})\phi_{\boldsymbol{m}}^\tau(\boldsymbol{x}) - f(\boldsymbol{x}) \right|.$$

For the first term, by Lemma 5.5 and A.23,

$$\left| \sum_{\boldsymbol{m} \in [M]^d} mult\left(\mathcal{P}_{\boldsymbol{m}}(\boldsymbol{x}), \mathcal{G}_{\boldsymbol{m}}(\boldsymbol{x})\right) - \sum_{\boldsymbol{m} \in [M]^d} \mathcal{P}_{\boldsymbol{m}}(\boldsymbol{x})\mathcal{G}_{\boldsymbol{m}}(\boldsymbol{x}) \right| \leq \frac{(M+1)^d}{12\lambda^2} \cdot \frac{\mathcal{P}_{\boldsymbol{m}}^4(\boldsymbol{x}) + 6\mathcal{P}_{\boldsymbol{m}}^2(\boldsymbol{x})\mathcal{G}_{\boldsymbol{m}}^2(\boldsymbol{x}) + \mathcal{G}_{\boldsymbol{m}}^4(\boldsymbol{x})}{8}$$

$$\leq \frac{(M+1)^d}{12\lambda^2} \cdot \frac{C_1^4 + 6C_1^2 C_2^2 + C_2^4}{8}.$$

For the second term, by Lemma A.17, A.22, A.1, and A.23,

$$\left| \sum_{\boldsymbol{m} \in [M]^d} \mathcal{P}_{\boldsymbol{m}}(\boldsymbol{x})\mathcal{G}_{\boldsymbol{m}}(\boldsymbol{x}) - \sum_{\boldsymbol{m} \in [M]^d} P_{\boldsymbol{m}}^\kappa(\boldsymbol{x})\phi_{\boldsymbol{m}}^\tau(\boldsymbol{x}) \right|$$

$$\leq \left| \sum_{\boldsymbol{m} \in [M]^d} \mathcal{P}_{\boldsymbol{m}}(\boldsymbol{x})\mathcal{G}_{\boldsymbol{m}}(\boldsymbol{x}) - \sum_{\boldsymbol{m} \in [M]^d} P_{\boldsymbol{m}}^\kappa(\boldsymbol{x})\mathcal{G}_{\boldsymbol{m}}(\boldsymbol{x}) \right| + \left| \sum_{\boldsymbol{m} \in [M]^d} P_{\boldsymbol{m}}^\kappa(\boldsymbol{x})\mathcal{G}_{\boldsymbol{m}}(\boldsymbol{x}) - \sum_{\boldsymbol{m} \in [M]^d} P_{\boldsymbol{m}}^\kappa(\boldsymbol{x})\phi_{\boldsymbol{m}}^\tau(\boldsymbol{x}) \right|$$

$$\leq \sum_{\boldsymbol{m} \in [M]^d} |\mathcal{G}_{\boldsymbol{m}}(\boldsymbol{x})| \cdot |\mathcal{P}_{\boldsymbol{m}}(\boldsymbol{x}) - P_{\boldsymbol{m}}^\kappa(\boldsymbol{x})| + \sum_{\boldsymbol{m} \in [M]^d} |P_{\boldsymbol{m}}^\kappa(\boldsymbol{x})| \cdot |\mathcal{G}_{\boldsymbol{m}}(\boldsymbol{x}) - \phi_{\boldsymbol{m}}^\tau(\boldsymbol{x})|$$

$$\leq \frac{(M+1)^d}{12\lambda^2} C_2 C_3 + \frac{(M+1)^d}{12\lambda^2} C_4(d-1).$$

For the third term, by Lemma 5.2,

$$\left| \sum_{\boldsymbol{m} \in [M]^d} P_{\boldsymbol{m}}^\kappa(\boldsymbol{x})\phi_{\boldsymbol{m}}^\tau(\boldsymbol{x}) - f(\boldsymbol{x}) \right| \leq \tau e^{-\tau} C_5 + M^{-\beta} C_6 + (M+1)^d \tau e^{-\tau} C_7.$$

$\square$

*Proof of Theorem 4.1.* Theorem A.24 directly implies Theorem 4.1 by setting

$$c_1 := 4 \left\lceil \frac{\kappa}{2} \right\rceil \binom{d+1}{d-1} + 12 \sum_{l=2}^{\kappa} \binom{d+l-1}{d-1}$$

$$c_2 := 24 + 16\lceil \log_2 d \rceil + 2d + 8 \left\lceil \frac{\kappa}{2} \right\rceil + 4 \sum_{l=2}^{\kappa} \binom{d+l-1}{d-1},$$

$$c_3 := \frac{\frac{C_1^4 + 6C_1^2 C_2^2 + C_2^4}{8} + C_2 C_3 + C_4(d-1)}{12},$$

$$c_4 := C_6,$$

$$c_5 := \frac{C_5}{2^d} + C_7.$$

$\square$

**Lemma A.25.** *For all $\tau \in \mathbb{R}$,*

$$e^{-\frac{\tau}{2}} \geq \tau e^{-\tau}. \tag{73}$$

*Proof of Lemma A.25.* Let $g(\tau) := e^{\frac{\tau}{2}} - \tau$, then the derivative $g'(\tau) = \frac{1}{2}e^{\frac{\tau}{2}} - 1$. Let $g'(\tau) = 0$, then the solution is $2\ln 2$. When $\tau \geq 2\ln 2$, $g'(\tau) \geq 0$; when $\tau \leq 2\ln 2$, $g'(\tau) \leq 0$. Therefore $g(\tau)$ decreases monotonically on $(-\infty, 2\ln 2]$, increases monotonically on $[2\ln 2, +\infty)$, and thus takes the minimum value $g(2\ln 2) = 2 - 2\ln 2 \geq 0$. It follows that

$$e^{\frac{\tau}{2}} - \tau \geq 0 \Rightarrow e^{\frac{\tau}{2}} \geq \tau \Rightarrow e^{-\frac{\tau}{2}} \geq \tau e^{-\tau}.$$

$\square$

*Proof of Corollary 4.2.* For all $0 < \varepsilon \leq 3c_4$, letting

$$M := \left\lceil 3^{\frac{1}{\beta}} c_4^{\frac{1}{\beta}} \varepsilon^{-\frac{1}{\beta}} \right\rceil = \mathcal{O}\left(\varepsilon^{-\frac{1}{\beta}}\right),$$

$$\lambda := \max\left\{ 2^{-\frac{1}{3}}, \sqrt{3^{\frac{d}{\beta}+d+1} c_3 c_4^{\frac{d}{\beta}} \varepsilon^{-\frac{\beta+d}{\beta}}} \right\} = \mathcal{O}\left(\varepsilon^{-\frac{\beta+d}{2\beta}}\right),$$

$$\tau := \max\left\{ 2\ln\left( 3^{\frac{d}{\beta}+d+1} c_4^{\frac{d}{\beta}} c_5 \varepsilon^{-\frac{\beta+d}{\beta}} \right), 1 \right\} = \mathcal{O}\left(\ln\frac{1}{\varepsilon}\right),$$

then

$$M \geq 3^{\frac{1}{\beta}} c_4^{\frac{1}{\beta}} \varepsilon^{-\frac{1}{\beta}} \geq 1,$$
$$\lambda \geq 2^{-\frac{1}{3}},$$
$$\tau \geq 1.$$

Therefore, by Theorem 4.1, because

$$(M+1)^d = \mathcal{O}\left(\varepsilon^{-\frac{d}{\beta}}\right),$$

the width of nonlinear layers, the width of linear hidden layers, and the number of nonzero parameters are $\mathcal{O}(\varepsilon^{-\frac{d}{\beta}})$; because

$$\lambda^2 = \mathcal{O}\left(\varepsilon^{-\frac{\beta+d}{\beta}}\right)$$

and

$$(3M+2)\tau = \mathcal{O}\left(\varepsilon^{-\frac{1}{\beta}} \ln\frac{1}{\varepsilon}\right),$$

the maximum absolute value of parameters is $\mathcal{O}(\varepsilon^{-\frac{\beta+d}{\beta}})$; because

$$c_3 \frac{(M+1)^d}{\lambda^2} \leq c_3 \frac{\left(3^{\frac{1}{\beta}} c_4^{\frac{1}{\beta}} \varepsilon^{-\frac{1}{\beta}} + 2\right)^d}{\lambda^2} \leq c_3 \frac{3^d 3^{\frac{d}{\beta}} c_4^{\frac{d}{\beta}} \varepsilon^{-\frac{d}{\beta}}}{3^{\frac{d}{\beta}+d+1} c_3 c_4^{\frac{d}{\beta}} \varepsilon^{-\frac{\beta+d}{\beta}}} = \frac{\varepsilon}{3},$$

$$c_4 M^{-\beta} \leq c_4 \left(3^{\frac{1}{\beta}} c_4^{\frac{1}{\beta}} \varepsilon^{-\frac{1}{\beta}}\right)^{-\beta} \leq \frac{\varepsilon}{3},$$

$$c_5 (M+1)^d \tau e^{-\tau} \leq c_5 (M+1)^d e^{-\frac{\tau}{2}} \leq c_5 \frac{3^d 3^{\frac{d}{\beta}} c_4^{\frac{d}{\beta}} \varepsilon^{-\frac{d}{\beta}}}{3^{\frac{d}{\beta}+d+1} c_4^{\frac{d}{\beta}} c_5 \varepsilon^{-\frac{\beta+d}{\beta}}} \leq \frac{\varepsilon}{3}, \qquad \text{(by Lemma A.25)}$$

the approximation error is no more than $\varepsilon$.

$\square$

# B. Supplementary Material for Experiments

*Table 2.* Basic information about UCI datasets used in experiments. The number of features for each dataset refers to the total number of features after converting categorical features into dummy features. For example, the categorical feature "race" in the dataset "Adult" takes values in the set {"White", "Asian-Pac-Islander", "Amer-Indian-Eskimo", "Other", "Black"} and we replace this feature with five dummy features taking values in {0, 1}.

| DATASET | UCI ID | # OBSERVATIONS | # FEATURES | TASK TYPE |
|---|---|---|---|---|
| Iris | 53 | 150 | 4 | Classification |
| Rice | 545 | 3810 | 7 | Classification |
| BankMarketing | 222 | 45211 | 47 | Classification |
| Adult | 2 | 48842 | 108 | Classification |
| RealEstate | 477 | 414 | 6 | Regression |
| Abalone | 1 | 4177 | 10 | Regression |
| WineQuality | 186 | 6497 | 13 | Regression |
| BikeSharing | 275 | 17379 | 12 | Regression |

