# OpenReview forum: "Approximation to Smooth Functions by Low-Rank Swish Networks"
_ICML.cc/2025/Conference — ICML 2025 poster_

### Official Review · Reviewer_r7Et · 2025-02-28

**Overall Recommendation:** 1

**Summary:**

The paper investigates whether deep neural networks that have been compressed using low‐rank factorization can still approximate smooth functions as accurately as uncompressed networks from a universal approximation theory perspective. In low‐rank compression, a weight matrix in a network layer is replaced by the product of two smaller matrices. This paper proposes a specific construction: between each pair of standard nonlinear (Swish) layers, a narrow linear layer without bias is inserted. This “bottleneck” enforces a low-rank structure on the transformation. The authors show, via a constructive proof, that with this constraint the network can approximate any function from the Holder space arbitrarily well. The construction uses classical tools from approximation theory. The domain is partitioned into a grid, and at each grid point the function is locally approximated by its Taylor polynomial. To control the approximation error away from the expansion point, the Taylor polynomials are “localized” using smooth bump functions that nearly vanish outside a small region. Both the Taylor polynomials and bump functions are then implemented using Swish networks.

**Claims And Evidence:**

The theoretical claims are well-supported.

**Essential References Not Discussed:**

The paper **"Approximation by Superpositions of a Sigmoidal Function"** should be cited. Published in 1989, it is one of the earliest works establishing universal approximation results for sigmoid networks.

**Experimental Designs Or Analyses:**

See above for my concerns regarding the experiments.

**Methods And Evaluation Criteria:**

For empirical evaluation, the authors conduct grid searches over network depth and width on eight UCI datasets. They compare the performance of a classical fully connected Swish network with a low-rank version, which uses narrow linear layers of one-third the width. The evaluation follows standard criteria: accuracy for classification and RMSE for regression. The use of 10-fold cross-validation and statistical tests adds rigor to their experimental validation. However, the tasks remain somewhat simplistic.

**Other Comments Or Suggestions:**

I am not convinced that universal approximation theory can be used to demonstrate the benefits of low-rank factorization. Moreover, the empirical results appear to be based on relatively simple tasks. At this stage, I find the paper unsuitable for acceptance.

**Other Strengths And Weaknesses:**

See below.

**Questions For Authors:**

No.

**Relation To Broader Scientific Literature:**

It is difficult to connect this paper to the broader scientific literature. I believe that universal approximation theory offers little practical insight and fails to convincingly demonstrate the advantage of the proposed low-rank structure.

**Theoretical Claims:**

The proofs have been checked loosely and appear to be correct.

---

> ### Author Rebuttal · Authors · 2025-03-25
>
> Thank you for your comments. We will try our best to relieve your concerns.
>
> Q1: The paper "Approximation by Superpositions of a Sigmoidal Function" should be cited.
>
> A1: Thank you for the reminder. We will add this groundbreaking work in our paper.
>
>
> Q2: I am not convinced that universal approximation theory can be used to demonstrate the benefits of low-rank factorization. Moreover, the empirical results appear to be based on relatively simple tasks.
>
> A2: Properly speaking, the aim of our work is not to demonstrate the benefits of low-rank compression. Low-rank compression, as a class of efficient and hardware-friendly network compression methods, can reduce the computation cost significantly, that is well examined in vast applications. However, as mentioned in our introduction: except low-rank compression, the remaining categories of methods are all underpinned by UATs to some extent. Hence, we aim to explain why low-rank networks can still perform comparably to full-rank ones from the perspective of approximation theory. Through our well-designed constructive approximation, we prove that the width of linear hidden layers needs only be no more than one-third of the width of nonlinear layers, implying that low-rank compression within this threshold does not impair the approximation capability of neural networks. By integrating our approximation results with sieve estimation theory, it is easy to derive the convergence rate of low-rank network estimates and limiting distributions for plug-in estimates based on these low-rank approximations.
>
> Regarding the simplicity of the experiments, our consideration is that the paper has already rigorously proven from the perspective of approximation theory that low-rank compression of weight matrices does not affect the approximation rate. The purpose of the experiments is to demonstrate the reliability of our theoretical proof. Just as one has used the axioms of Euclidean geometry to prove that the base angles of any isosceles triangle are equal, there is no subsequent need to draw numerous different isosceles triangles and measure their base angles for verification. According to academic conventions in neural network approximation theory, the vast majority of papers—including the work “Approximation by Superpositions of a Sigmoidal Function”—do not include any experimental validation.
>
>
> Thank you once again. If there are still any doubts, we sincerely hope that you will kindly offer your comments.

---

### Official Review · Reviewer_US5n · 2025-03-08

**Overall Recommendation:** 2

**Summary:**

This paper discusses the universal approximation property of row-rank MLPs using the Swish activation function.

## update after rebuttal
Although the authors provide some explanation regarding the activation function and the number of parameters, my concerns are not fully addressed. Therefore, I will maintain my score.

**Claims And Evidence:**

The authors provide the number of parameters required for the approximation used in Theorem 4.1, along with a bound. While I have not rigorously checked every detail of the proof, I find no issues with its overall structure.

**Essential References Not Discussed:**

.

**Experimental Designs Or Analyses:**

.

**Methods And Evaluation Criteria:**

The authors approximated the function using a commonly employed method: partitioning the space and connecting each partition with a bump function.

**Other Comments Or Suggestions:**

The authors mention that low-rank approximation is used as one of four methods for compressing neural networks. The paper does cite prior works related to low-rank approximation, but it does not explicitly specify the source of the definition of low-rank MLP when introducing it in Section 3.2. Clearly stating where this definition originates from would enhance clarity and strengthen the paper’s argument.

**Other Strengths And Weaknesses:**

The proof technique is highly standard, and it is difficult to consider it a significant mathematical advancement.

The choice of activation function is too restrictive. Why is the activation function in the paper restricted to Swish? As I understand it, the properties of Swish are used in Lemmas 5.4–5.6 to approximate $xy$, $x^2$, and $x$, but these approximations can also be achieved using many other activation functions. Additionally, approximating bump functions should also be feasible with alternative activations.

I do not quite understand the significance of the linear hidden layer having a smaller width than the nonlinear layer. These values merely compare the number of parameters within the same network, whereas a meaningful comparison should be made between different networks.

The authors conducted a quantitative approximation comparison only with Ohn & Kim (2019), but they should also compare their results with more recent studies for a more comprehensive evaluation.

**Questions For Authors:**

.

**Relation To Broader Scientific Literature:**

.

**Theoretical Claims:**

The main structure of the proof follows a natural progression, and I do not anticipate any major issues.

---

> ### Author Rebuttal · Authors · 2025-03-26
>
> Thank you for your comments. We will try our best to relieve your concerns.
>
> Q1: The proof technique is highly standard, and it is difficult to consider it a significant mathematical advancement.
>
> A1: While our mathematical proof follows a well-established framework, we conducted a well-designed constructive approximation which guarantees that the width of linear hidden layers is no more than one-third of the width of nonlinear layers. To our knowledge, though many works demonstrated that low-rank compression performs in a variety of applications empirically, we are the first to develop the theoretical foundation for low-rank compression from the perspective of approximation theory by answering the question of whether a neural network processed by low-rank compression can serve as a good approximator for a wide range of functions. We believe that mathematics is a tool for reaching truth and what matters is the conclusion derived from the mathematical tools.
>
> Q2: The choice of activation function is too restrictive.
>
> A2: From the perspective of technical proofs, we approximated the identity function precisely by leveraging the characteristics of swish function (Lemma 5.6). From the perspective of foundational contributions to future research, as highlighted in the Introduction, the infinite differentiability of Swish neural networks creates unique opportunities to extend our approximation theorems to higher-order Sobolev spaces.
>
>
> Q3: I do not quite understand the significance of the linear hidden layer having a smaller width than the nonlinear layer.
>
> A3: As mentioned in the Introduction, to our knowledge, there is no work providing a theorical explanation to answer why low-rank compression does not hurt performance of neural networks. We are the first one to develop the theoretical foundation for low-rank compression from the perspective of approximation theory by answering the question of whether a neural network processed by low-rank compression can serve as a good approximator for a wide range of functions. Though our compression rate is not very significant, we believe that our publication will inspire more important theories.
>
>
> Q4: The authors conducted a quantitative approximation comparison only with Ohn & Kim (2019), but they should also compare their results with more recent studies for a more comprehensive evaluation.
>
> A4: While there are many articles discussing the approximation capabilities of neural networks, most focus on ReLU neural networks. Additionally, most studies target function classes of continuous functions, Lipschitz functions, and Hölder continuous functions. Many works also do not provide simultaneously the required depth, width, upper bound of absolute values of parameters, and upper bound of number of nonzero parameters for approximation. Considering all the above points, we selected Ohn & Kim (2019) for comparison.
>
>
> Q5: it does not explicitly specify the source of the definition of low-rank MLP when introducing it in Section 3.2.
>
> A5: Up to now, there is no mathematically rigorous and universally accepted definition of low-rank networks. After all, even the strict mathematical definition of neural networks themselves lacks a consensus—how can we expect one for low-rank networks? Our Definition 3.1 builds upon extensive prior research on low-rank compression, proposing a definition aligned with the majority of scholars' understanding. This definition captures the essence of low-rank compression: reducing computational cost by decomposing large matrices (or convolutional kernels) into products of smaller matrices (or smaller convolutional kernels). We will supplement additional references to demonstrate that this definition aligns with the network architectures proposed in these studies:
>
> [1]. Denil, Misha, et al. "Predicting parameters in deep learning." Advances in neural information processing systems 26 (2013).
>
> [2]. Idelbayev, Yerlan, and Miguel A. Carreira-Perpinán. "Low-rank compression of neural nets: Learning the rank of each layer." Proceedings of the IEEE/CVF conference on computer vision and pattern recognition. 2020.
>
> [3]. Sainath, Tara N., et al. "Low-rank matrix factorization for deep neural network training with high-dimensional output targets." 2013 IEEE international conference on acoustics, speech and signal processing. IEEE, 2013.
>
>
> Thank you once again. If there are still any doubts, we sincerely hope that you will kindly offer your comments.

---

### Official Review · Reviewer_j7PK · 2025-03-17

**Overall Recommendation:** 2

**Summary:**

This paper investigates low-rank compression techniques for neural networks by strategically inserting narrow linear layers between adjacent nonlinear layers. The authors theoretically demonstrate that low-rank Swish networks with a fixed depth can approximate any function within a Hölder ball Cβ,R([0,1]^d) to an arbitrarily small error. Notably, the theoretical analysis establishes bounds on the minimal width required for these inserted linear layers, achieving substantial reductions in computational cost without significant accuracy loss.

**Claims And Evidence:**

I can understand the claims in Section3, but not much in Section 4 and 5.

**Essential References Not Discussed:**

No.

**Experimental Designs Or Analyses:**

Please see the evaluation part for my confusion of model and dataset selection.

**Methods And Evaluation Criteria:**

The selecting specific datasets and models in the evaluation is unclear. Clarification on the choice and potential applicability to more prevalent and larger-scale models would significantly strengthen the practical impact of the method.

**Other Comments Or Suggestions:**

See strengths and weaknesses.

**Other Strengths And Weaknesses:**

Strengths:

Rigorous Mathematical Proofs: The paper provides extensive mathematical proofs, rigorously demonstrating the proposed theorem, significantly contributing to the theoretical understanding of low-rank compression.

Clear Compression Theory: The authors articulate a clear theoretical basis for the achievable compression ratio and computational reductions, providing a solid analytical foundation.

Innovative Use of Hölder Ball: Employing the Hölder ball in the analysis introduces a novel perspective, enriching the theoretical contributions of the paper.

Weaknesses:

Complexity and Readability of Proof: The mathematical proofs, while rigorous, are challenging to follow due to the absence of a clearly structured narrative to guide readers effectively.

Lack of Runtime Performance Analysis: The paper lacks empirical data on real-world runtime reductions. Providing measurements of actual runtime improvements could significantly validate the practical utility of the methods.

**Questions For Authors:**

See strengths and weaknesses.

**Relation To Broader Scientific Literature:**

This paper offers valuable theoretical insights into structured low-rank compression strategies.

**Theoretical Claims:**

The theory and proofs in section 4 and 5 are hard to follow for me.

---

> ### Author Rebuttal · Authors · 2025-03-27
>
> Thank you for your comments. We will try our best to relieve your concerns.
>
> Q1: Complexity and Readability of Proof
>
> A1: To establish a novel theoretical foundation for low-rank compression, we develop a rigorous mathematical framework comprising multiple interlocking proofs that, while intricate for researchers new to neural network approximation theory, are indispensable for upholding formal rigor. To enhance accessibility, we distill the conceptual underpinnings of our constructive approximation framework in a dedicated subsection at the conclusion of the Introduction (preceding the contributions summary). In Section 5, we adopt a modular approach to present the complete derivation: Theorem 4.1 is decomposed into four self-contained modules, each addressing distinct aspects of the proof architecture while highlighting critical technical innovations and their interrelationships.
>
> Q2: Lack of Runtime Performance Analysis
>
> A2: The purpose of this paper is not to prove how much low-rank compression, as a network compression method, can reduce neural network inference time. Low-rank compression has been proven in many literatures to be an efficient, easy-to-implement, and hardware-friendly network compression method that significantly reduces computational cost. As emphasized in the Introduction, our work aims to develop a theoretical foundation for low-rank compression from the perspective of approximation theory. Our Theorem 4.1 provides an upper bound of approximation error for representing any function in the Hölder ball $\mathcal{C}^{\beta,R}([0,1]^d)$ using low-rank Swish networks. Consequently, our experiments also focus on error analysis. Furthermore, Section 4 presents theoretical calculations confirming that the low-rank architectures in Theorem 4.1 reduce the number of multiplication operations—a more scientifically rigorous metric than runtime, as runtime is highly susceptible to environmental variability.
>
>
> Thank you once again. If there are still any doubts, we sincerely hope that you will kindly offer your comments.

---

### Official Review · Reviewer_uBR2 · 2025-03-22

**Overall Recommendation:** 3

**Summary:**

This paper studies the question of whether networks with low rank matrices and Swish activations can approximate a class of Holder-continuous and smooth functions. The authors show that the number of parameters and operations can be reduced by 1/3 to still obtain the same approximation rates.

**Claims And Evidence:**

The theoretical claims are supported by proofs. While I have not checked the proofs thoroughly, the logical structure of the proofs seems to be correct. The empirical evaluations in section 6 demonstrate comparable performance between low rank networks and full rank networks, however these are on small scale datasets and with small networks. Nevertheless the literature contains substantial empirical support for the natural emergence of low-rank structures within neural network layers. Since the main contribution of the paper is the theoretical results, I will not dwell on the empirical evaluation.

**Essential References Not Discussed:**

I am not sure.

**Experimental Designs Or Analyses:**

Some more details about the training (choice of regularization, optimizer, other hyperparameters) could be provided.

**Methods And Evaluation Criteria:**

Yes

**Other Comments Or Suggestions:**

None.

**Other Strengths And Weaknesses:**

Weaknesses: The results do not seem to escape the curse of dimensionality. One hope of using low rank layers is to achieve more efficient approximation rates. This does not seem to be the case here.

The improvement in terms of number of parameters/computational steps is only a constant 2/3 factor.

The role played by the Swish activation is unclear - why is that a necessary component of the network architecture?

Overall, while the paper proves that networks with Swish activations and low rank layers can approximate Holder functions at the same rate, the reasons to prefer these Low rank swish networks are not very compelling.

The amount of savings that can typically be achieved in practice is much more than 33%, so it is not clear the current results can explain how to achieve better efficiency.

**Questions For Authors:**

Is there an advantage to using Low rank swish networks over full rank ReLU networks in terms of approximation? Or is your goal to show that Low rank swish networks achieve comparable rates of approximation?

**Relation To Broader Scientific Literature:**

The approximation results of the paper are in line with other neural network approximation results like [1] (for Sobolev spaces, not Holder).


[1] Hrushikesh N Mhaskar. Neural networks for optimal approximation of smooth and analytic functions. Neural computation, 8(1):164–177, 1996

**Theoretical Claims:**

I did not check the correctness of the proofs in detail. However the results and proofs in the paper seem similar to other results on approximation with neural networks so I trust them.

---

> ### Author Rebuttal · Authors · 2025-03-27
>
> Thank you for your comments. We will try our best to relieve your concerns.
>
> Q1: Problem on the curse of dimensionality.
>
> A1: If you want to approximate any function in a Hölder ball, then theoretically it is impossible to escape the curse of dimensionality[1, 2]. In Remark 4.5 of Section 4, we introduce a novel function class that is prevalent in practical applications, and when approximating functions within this class, one can avoid the curse of dimensionality.
>
> Q2: Problem on the compression rate.
>
> A2: As mentioned in the Introduction, to our knowledge, there is no work providing a theorical explanation to answer why low-rank compression does not hurt performance of neural networks. We are the first one to develop the theoretical foundation for low-rank compression from the perspective of approximation theory by answering the question of whether a neural network processed by low-rank compression can serve as a good approximator for a wide range of functions. Though our compression rate is not very significant, we believe that our publication will inspire more important theories.
>
> Q3: Problem on the choice of activation function
>
> A3: From the perspective of technical proofs, we approximated the identity function precisely by leveraging the characteristics of swish function (Lemma 5.6). From the perspective of foundational contributions to future research, as highlighted in the Introduction, the infinite differentiability of Swish neural networks creates unique opportunities to extend our approximation theorems to higher-order Sobolev spaces.
>
> Thank you once again. If there are still any doubts, we sincerely hope that you will kindly offer your comments.
>
>
> [1]. Schmidt-Hieber, A. J. Nonparametric Regression using Deep Neural Networks with ReLU Activation Function. The Annals of statistics, 48(4):1875–1897, 2020.
>
> [2]. Bauer, B. and Kohler, M. On deep learning as a remedy for the curse of dimensionality in nonparametric regression. The Annals of statistics, 47(4):2261 - 2285, 2019.

---

### Decision · Program_Chairs · 2025-05-01

**Decision:**

Accept (poster)

**Comment:**

This paper examines whether networks using low-rank matrices and Swish activations can effectively approximate Holder-continuous and smooth functions, assuming a certain scaling of the width of the hidden layers. The question is interesting and relevant to neural network theory (in particular, approximation theory). Some reviewers doubted relevance of approximation-theoretic analysis in the context of low-rank learning, while others found that the proof appears to follow fairly standard steps. Perhaps an interesting point in such a context is whether low-rank compression manages to mitigate the curse of dimensionality, to which paper shows that in some classes this is indeed possible (Sec. 4).